# Unsupervised Point Cloud Completion through Unbalanced Optimal Transport

## Abstract

Unpaired point cloud completion explores methods for learning a completion map from unpaired incomplete and complete point cloud data. In this paper, we propose a novel approach for unpaired point cloud completion using the unbalanced optimal transport map, called Unbalanced Optimal Transport Map for Unpaired Point Cloud Completion (UOT-UPC). We demonstrate that the unpaired point cloud completion can be naturally interpreted as the Optimal Transport (OT) problem and introduce the Unbalanced Optimal Transport (UOT) approach to address the class imbalance problem, which is prevalent in unpaired point cloud completion datasets. Moreover, we analyze the appropriate cost function for unpaired completion tasks. This analysis shows that the InfoCD cost function is particularly well-suited for this task. Our model is the first attempt to leverage UOT for unpaired point cloud completion, achieving competitive or superior results on both single-category and multi-category datasets. In particular, our model is especially effective in scenarios with class imbalance, where the proportions of categories are different between the incomplete and complete point cloud datasets.

## 1 Introduction

The three-dimensional (3D) point cloud is a fundamental representation of 3D geometry processing (Guo et al., 2020). However, obtaining complete point cloud data is challenging because of the limitations of the scanning process (Yuan et al., 2018). In this respect, many methods have been proposed for point cloud completion, which aims to recover a complete point cloud from incomplete (partial) data (Yu et al., 2021; Wang et al., 2022; Tchapmi et al., 2019; Chen et al., 2020; Hong et al., 2023). These previous approaches can be categorized into paired (supervised) and unpaired (unsupervised) methods. In the paired approach, the completion model is trained using paired data, which consists of incomplete point clouds and their corresponding completions (Yu et al., 2021; Wang et al., 2022; Tchapmi et al., 2019; Xia et al., 2021; Zhou et al., 2021). However, acquiring this paired training data is often difficult in practice. Therefore, the unpaired point cloud completion aims to train a completion model from the independently sampled incomplete and complete point clouds, leveraging shared semantic information, such as object class (Ma et al., 2023; Chen et al., 2020; Wen et al., 2021), or through domain adaptation using paired synthetic data (Liu et al., 2024).. In this regard, the unpaired point cloud completion is a challenging task of significant practical importance.

Optimal Transport problem (OT) problem (Villani et al., 2009; Peyré et al., 2017) investigates the cost-minimizing transport map that bridges two probability distributions. Since the introduction of WGAN (Arjovsky et al., 2017), the OT-based Wasserstein distance has been widely adopted as a loss function in various machine learning tasks, including unpaired point cloud completion (Chen et al., 2020; Wu et al., 2020). Recently, several works introduced alternative approaches based on OT (Rout et al., 2022; Fan et al., 2022). Instead of estimating the Wasserstein distance, these works focus on learning the optimal transport map (**OT Map**) from the source distribution to the target distribution using neural networks. Intuitively, the optimal transport map $T$ serves as a generator of the target distributions which minimizes the given cost function. In this respect, **this cost function plays a crucial role for $T$, because it determines how each input $x$ is transported to $T(x)$.**

In this paper, we introduce a novel unpaired point cloud completion model based on the unbalanced optimal transport map. We refer to our model as the ***Unbalanced Optimal Transport Map for Unpaird Point Cloud Completion (UOT-UPC)***. We formulate the unpaired point cloud completion

task as the optimal transport problem and investigate the suitable cost function for this task. Note that the completion model is required to generate the correct complete point cloud corresponding to each incomplete point cloud, not an arbitrary complete one. Therefore, identifying the proper cost function is crucial for UOT-UPC. Moreover, we demonstrate that the class imbalance problem exists in unpaired point cloud completion. Then, we verify that the Unbalanced Optimal Transport (UOT) framework presents favorable properties for addressing this class imbalance. Our experiments demonstrate that UOT-UPC achieves state-of-the-art performance in unpaired point cloud completion in both single-category and multi-category settings. Furthermore, UOT-UPC exhibits particularly robust performance when handling the class imbalance. Our contributions are summarized as follows:

- To the best of our knowledge, UOT-UPC is the first unpaired point cloud completion model based on the Unbalanced Optimal Transport map.

- We formulate unpaired point cloud completion as the task of finding the optimal transport map (OT Map) and analyze the most suitable transport cost function for this task.

- UOT-UPC attains state-of-the-art performance in unpaired point cloud completion in both single-category and multi-category settings.

- We demonstrate that UOT-UPC exhibits significant robustness to class imbalance. This robustness is induced by its UOT formulation.

**Notations and Assumptions**  Let $\mathcal{X}$, $\mathcal{Y}$ be two compact complete metric spaces, $\mu$ and $\nu$ be probability distributions on $\mathcal{X}$ and $\mathcal{Y}$, respectively. $\mu$ and $\nu$ are assumed to be absolutely continuous with respect to the Lebesgue measure. Throughout this paper, we denote the source distribution as $\mu$ and the target distribution as $\nu$. Since the focus of this paper is on point cloud completion, $\mu$ **and $\nu$ represent the distributions of the incomplete and complete point clouds**, respectively. For a measurable map $T$, $T_{\#}\mu$ represents the pushforward distribution of $\mu$. $\Pi(\mu,\nu)$ denote the set of joint probability distributions on $\mathcal{X} \times \mathcal{Y}$ whose marginals are $\mu$ and $\nu$, respectively. Additionally, $f^*$ indicates the convex conjugate of a function $f$, i.e., $f^*(y) = \sup_{x \in \mathbb{R}}\{\langle x, y \rangle - f(x)\}$ for $f : \mathbb{R} \to [-\infty, \infty]$.

## 2 BACKGROUND

**Optimal Transport**  The *Optimal Transport (OT)* problem investigates the task of transporting the source distribution $\mu \in \mathcal{P}(\mathcal{X})$ to the target distribution $\nu \in \mathcal{P}(\mathcal{Y})$. This problem was initially formulated by Monge (1781) using a deterministic transport map $T : \mathcal{X} \to \mathcal{Y}$ such that $T_{\#}\mu = \nu$:

$$C_{ot}(\mu,\nu) := \inf_{T_{\#}\mu=\nu} \left[ \int_{\mathcal{X}} c(x, T(x))d\mu(x) \right]. \tag{1}$$

Intuitively, Monge's OT problem explores the optimal transport map $T^*$ that connects two distributions while minimizing the given cost function $c(x, T(x))$. Although Monge's OT problem offers an intuitive understanding, it has theoretical limitations: this formulation is non-convex and the optimal transport map $T^*$ may not exist depending on the conditions on $\mu$ and $\nu$ (Villani et al., 2009). To overcome these issues, Kantorovich introduced a relaxed formulation of the OT problem (Kantorovich, 1948). Formally, this Kantorovich formulation is expressed in terms of a coupling $\pi$ rather than a transport map $T$, as follows:

$$C_{ot}(\mu,\nu) := \inf_{\pi \in \Pi(\mu,\nu)} \left[ \int_{\mathcal{X} \times \mathcal{Y}} c(x, y)d\pi(x, y) \right]. \tag{2}$$

where $c$ is a cost function and $\pi \in \Pi(\mu,\nu)$ is a coupling of $\mu$ and $\nu$. In contrast to the Monge problem, the minimizer $\pi^\star$ of Eq 2 always exists under some mild assumptions on $(\mathcal{X},\mu)$, $(\mathcal{Y},\nu)$ and the cost function $c$ (Villani et al., 2009). Note that under our assumptions that $\mu$ and $\nu$ are absolutely continuous with respect to the Lebesgue measure, the deterministic optimal transport map $T^*$ exists and the optimal coupling is given by $\pi^\star = (Id \times T^\star)_{\#}\mu$ (Villani et al., 2009).

Rout et al. (2022); Fan et al. (2022) proposed a method for learning the optimal transport map $T^\star$ using the semi-dual formulation of OT. This neural network-based approach for learning the optimal transport map is referred to as *Neural Optimal Transport (Neural OT)*. These works applied

Neural OT to generative modeling and image-to-image translation tasks. In specific, these models parametrize the potential function $v$ and the transport map $T$ as follows:

$$\mathcal{L}_{v_\phi, T_\theta} = \sup_{v_\phi} \left[ \int_{\mathcal{X}} \inf_{T_\theta} \left[ c\left(x, T_\theta(x)\right) - v_\phi\left(T_\theta(x)\right) \right] d\mu(x) + \int_{\mathcal{X}} v_\phi(y) d\nu(y) \right]. \tag{3}$$

**Unbalanced Optimal Transport**   The classical OT problem assumes an exact transport between two distributions $\mu$ and $\nu$, i.e., $\pi_0 = \mu, \pi_1 = \nu$. However, this exact matching constraint results in sensitivity to outliers (Balaji et al., 2020; Séjourné et al., 2022) and vulnerability to class imbalance in the classical OT problem (Eyring et al., 2024). To mitigate this issue, a new variation of the optimal transport problem is introduced, called *Unbalanced Optimal Transport (UOT)* (Chizat et al., 2018; Liero et al., 2018). Formally, the UOT problem is expressed as follows:

$$C_{uot}(\mu, \nu) = \inf_{\pi \in \mathcal{M}_+(\mathcal{X} \times \mathcal{Y})} \left[ \int_{\mathcal{X} \times \mathcal{Y}} c(x, y) d\pi(x, y) + D_{\Psi_1}(\pi_0 | \mu) + D_{\Psi_2}(\pi_1 | \nu) \right], \tag{4}$$

where $\mathcal{M}_+(\mathcal{X} \times \mathcal{Y})$ denotes the set of positive Radon measures on $\mathcal{X} \times \mathcal{Y}$. $D_{\Psi_1}$ and $D_{\Psi_2}$ represents two $f$-divergences generated by convex functions $\Psi_i$, and are defined as $D_{\Psi_i}(\pi_i | \eta) = \int \Psi_i \left( \frac{d\pi_i(x)}{d\eta(x)} \right) d\eta(x)$. These $f$-divergences penalize the discrepancies between the marginal distributions $\pi_0, \pi_1$ and $\mu, \nu$, respectively. Hence, **in the UOT problem, the two marginal distributions are softly matched to** $\mu, \nu$, i.e., $\pi_0 \approx \mu$ and $\pi_1 \approx \nu$. Intuitively, the UOT problem can be seen as the OT problem between $\pi_0 \approx \mu$ and $\pi_1 \approx \nu$, rather than between the exact distributions $\mu$ and $\nu$ (Choi et al., 2023). This flexibility offers robustness to outliers (Balaji et al., 2020) and adaptability to class imbalance problem between $\mu$ and $\nu$ (Eyring et al., 2024) to the UOT problem (See Sec 3.2 for details). We refer to the optimal transport map $T^\star$ from $\pi_0$ to $\pi_1$ as the ***unbalanced optimal transport map***.

Choi et al. (2023) introduced a Neural OT model for the UOT problem into generative modeling, called UOTM (See Sec 3.2 for details). In this paper, we introduce the unbalanced optimal transport map to unpaired point cloud completion. Unlike generative modeling, in unpaired point cloud completion, each incomplete source sample $x$ should be transported to its corresponding complete target sample $y$. Therefore, it is important to set an appropriate cost function $c(x, y)$ in Eq 4, because this cost determines how each $x$ is transported to $y$ in the optimal transport map. In Sec 3.1, we investigate the optimal cost function for unpaired point cloud completion.

## 3   UNPAIRED POINT COMPLETION THROUGH UNBALANCED OPTIMAL TRANSPORT MAP

In this paper, our key idea is to **train our model to learn the unbalanced optimal transport map from the incomplete point cloud distribution** $\mu$ **to the complete point cloud distribution** $\nu$. In Sec 3.1, we demonstrate that this optimal transport approach is appropriate for the unpaired point cloud completion task. In particular, we investigate the most appropriate cost function for this application. In Sec 3.2, we present our max-min learning objective. In Sec 3.3, we provide implementation details, such as neural network parametrization and training algorithm.

### 3.1   MOTIVATION

**Task Formulation as Optimal Transport Map**   We begin by formulating our target task: *Unpaired point cloud completion*. Assume that we are given two sets of point cloud data: the incomplete point cloud $X = \{x_i \mid x_i \in \mathcal{X}, i = 1, \cdots, N\}$ and the complete point cloud $Y = \{y_j \mid y_j \in \mathcal{Y}, j = 1, \cdots, M\}$. Note that $X$ and $Y$ are not paired, i.e., $X$ and $Y$ are independently sampled from the incomplete point cloud distribution $\mu$ and the complete point cloud distribution $\nu$, respectively. In practice, obtaining complete point clouds for real-world scene data is often prohibitively expensive, making this unsupervised approach essential (Ma et al., 2023). Formally, our goal is to train a point completion model $T$ using the unpaired datasets:

$$T : \mathcal{X} \to \mathcal{Y}, \quad x \text{ (Incomplete point cloud)} \mapsto T(x) \text{ (Point cloud completion)}. \tag{5}$$

This point cloud completion model $T$ must satisfy the following two conditions.

Figure 1: **Visualization of the incomplete point cloud $x$, the ground-truth completion $y^{gt}(x)$, and the three complete point clouds $y^c_i(x)$** that minimize the cost $c(x, y^c_i(x)$ for two cost functions: $cd^{l2}$ and InfoCD, in the **multi-category setting**.

Table 1: **Comparison between the cost-minimizer $g^c_1(x)$ and the ground-truth completion $y^{gt}(x)$ for each incomplete point cloud $x$ across diverse cost function $c(\cdot, \cdot)$.** We evaluate the optimality of each cost function by measuring the L1 Chamfer distance $cd^{l1} \times 10^2 (\downarrow)$ between $g^c_1(x)$ and $y^{gt}(x)$.

(a) Single-category

| Cost Function | AVG | chair | table | trash bin | TV | cabinet | bookshelf | sofa | lamp | bed | tub |
|---|---|---|---|---|---|---|---|---|---|---|---|
| USSPA | 7.18 | 7.44 | 7.15 | 6.98 | 6.08 | 10.02 | 7.00 | 6.12 | 8.35 | 7.90 | 4.79 |
| $l_2$ | 14.88 | 11.21 | 12.52 | 22.37 | 8.29 | 20.46 | 17.87 | 8.69 | 11.57 | 19.55 | 7.07 |
| $cd^{l2}$ | 6.65 | 7.17 | 7.35 | 8.35 | 5.46 | 10.59 | 5.77 | 6.39 | 3.70 | 6.46 | 5.28 |
| $cd^{l2}_{fwd}$ | 6.12 | 7.29 | 7.41 | 7.23 | **5.18** | 9.03 | 6.45 | **4.64** | 2.82 | 6.75 | **4.44** |
| InfoCD | **5.58** | **6.84** | **5.90** | **6.91** | 5.29 | **7.86** | **4.37** | 5.75 | **2.72** | **5.78** | 4.51 |

(b) Multi-category

| Cost Function | AVG | chair | table | trash bin | TV | cabinet | bookshelf | sofa | lamp | bed | tub |
|---|---|---|---|---|---|---|---|---|---|---|---|
| USSPA | 8.64 | **7.40** | 8.88 | **9.13** | 8.70 | 11.48 | 7.61 | 6.52 | 10.01 | 8.72 | 8.30 |
| $l_2$ | 23.97 | 12.52 | 31.21 | 29.17 | 26.65 | 22.29 | 22.96 | 20.51 | 24.64 | 27.03 | 21.80 |
| $cd^{l2}$ | 9.78 | 8.07 | 7.69 | 14.00 | 5.91 | 18.86 | 7.88 | 7.34 | 6.23 | 8.76 | 7.07 |
| $cd^{l2}_{fwd}$ | 8.87 | 9.48 | 8.62 | 9.38 | 7.80 | **10.55** | 7.73 | **5.63** | 14.59 | 10.32 | 7.28 |
| InfoCD | **8.46** | 7.43 | **6.41** | 11.69 | **5.69** | 17.35 | **6.52** | 6.25 | **2.70** | **6.91** | **4.92** |

(i) $T$ should generate a complete point cloud sample, i.e., $y = T(x) \sim \nu$.

(ii) $T$ should transport each incomplete point cloud to its corresponding complete point cloud $y$, rather than to any arbitrary complete point cloud.

In this regard, the optimal transport map (Eq. 1) is suitable for the point completion model. By definition, the optimal transport map $T^\star$ is (1) a generator of the complete point cloud samples, i.e., $T(x) \sim \nu$ for $x \sim \mu$ that (2) optimally minimizes the given cost function $c(x, T(x))$. Thus, the first condition (i) is naturally satisfied. **If we can identify a suitable cost function $c(\cdot, \cdot)$ that induces an explicit bias in $T^\star$ to satisfy (ii), then $T^\star$ can serve as the point cloud completion model. Specifically, this suitable cost function $c(\cdot, \cdot)$ should assign a lower cost to $c(x, T(x))$ when $T(x)$ is the correct completion of $x$ and a higher cost to $c(x, y)$ when $y$ is not the correct corresponding completion.**

**Cost Function Comparison** We conducted the following experiments to evaluate whether the cost-minimizing pair of each cost function is appropriate for the unpaired point cloud completion tasks. We test various cost function candidates, including $l_2$, $L2$-Chamfer distance ($cd^{l2}$) (Fan et al., 2017), one-directional $L2$-Chamfer distance ($cd^{l2}_{fwd}$), and InfoCD (Lin et al., 2024). Each cost

Table 2: **Class imbalance in the benchmark dataset from (Ma et al., 2023).** The Incomplete and Complete rows indicate the proportion of each class in the respective datasets. The Ratio represents the proportion ratio (incomplete/complete). A Ratio $\neq 1$ indicates the presence of class imbalance.

| class | chair | table | trash bin | TV | cabinet | bookshelf | sofa | lamp | bed | tub |
|---|---|---|---|---|---|---|---|---|---|---|
| Incomplete | 43% | 21.3% | 8.0% | 6.4% | 6.0% | 6.1% | 3.9% | 1.1% | 2.9% | 1.2% |
| Complete | 22.2% | 22.2% | 1.9% | 6.1% | 8.7% | 2.5% | 17.6% | 12.9% | 1.3% | 4.7% |
| Ratio | 1.94 | 0.96 | 4.21 | 1.05 | 0.69 | 2.44 | 0.22 | 0.09 | 2.23 | 0.26 |

function is defined as follows for an incomplete (partial) point cloud $x_i = \{x_{im} \in \mathbb{R}^3\}$ and complete point cloud $y_i = \{y_{in} \in \mathbb{R}^3\}$.

- $l_2(x_i, y_j) = \sum_m \|x_{im} - y_{im}\|_2^2$.
- $cd^{l2}(x_i, y_j) = \sum_m \min_n \|x_{im} - y_{in}\|_2^2 + \sum_n \min_m \|x_{im} - y_{in}\|_2^2$.
- $cd^{l2}_{fwd}(x_i, y_j) = \sum_m \min_n \|x_{im} - y_{in}\|_2^2$.
- $\text{InfoCD}(x_i, y_j) = \ell_{\text{InfoCD}}(x_i, y_j) + \ell_{\text{InfoCD}}(y_j, x_i)$.
  where $\ell_{\text{InfoCD}}(x_i, y_i) = -\frac{1}{|y_i|} \sum_n \log \left\{ \frac{\exp\{-\frac{1}{\tau'} \min_m d(x_{im}, y_{in})\}}{\sum_n \exp\{-\frac{1}{\tau} \min_m d(x_{im}, y_{in})\}} \right\}$

For each partial point cloud $x$ and a given cost function $c(\cdot, \cdot)$, we select $k$-nearest complete samples $y_i^c(x)$ for $1 \leq i \leq k$ based on $c(x, \cdot)$ on the target dataset. Then, we compare them with the ground-truth completion $y^{gt}(x)$. Our goal is to evaluate each cost function by testing whether the $k$-nearest neighbor $y_i^c(x)$ is indeed similar to the ground-truth completion $y^{gt}(x)$. If so, this suitable cost function can be exploited to train our OT-based completion model via the optimal transport map. The experiment is conducted on paired completion data from ShapeNet (Chang et al., 2015). In the single-category setting, $y_i^c(x)$ is selected from the set of ground-truth completions within the same category. In the multi-category setting, $y_i^c(x)$ is selected from a mixture of ground-truth completions from the ten categories, such as chairs, tables, trash bins, etc. For comparison, we also trained and evaluated the competitive USSPA model (Ma et al., 2023) on each dataset.

Fig. 1 visualize the incomplete point cloud $x$, the ground-truth completion $y^{gt}(x)$, and the 3-nearest neighbor $y_3^c(x)$ for the $cd^{l2}$ and InfoCD cost functions. Fig. 1 show that selecting the cost-minimizing pair based on InfoCD retrieves an appropriate $y_3^c(x)$, which closely resembles $y^{gt}(x)$, in the multi-category setting (See Appendix B for additional results for other cost functions and the single-category setting). Table 1 presents similar results. Table 1 reports the L1 chamfer distance between $y^{gt}(x)$ and the nearest neighbor $y_1^c(x)$ for each cost function. The results indicate that the $l2$ cost performs the worst. This result shows that $l_2$ cost is unsuitable for the point cloud completion task. In contrast, the InfoCD achieves competitive results, performing comparably or better than USSPA on the majority of datasets. **Therefore, in Sec 3.2, we propose an OT Map approach using the InfoCD cost function for the point cloud completion task, based on our investigation of the most suitable cost function.** Furthermore, we conduct an ablation study on the cost function in Sec 5.3 to demonstrate how this cost function comparison closely aligns with the completion performance of UOT-UPC.

**Unbalanced Optimal Transport Map for Class Imbalance Problem**   In this paragraph, we clarify the motivation for considering the unbalanced optimal transport map, instead of the classical optimal transport map. Our goal in this paper is unpaired point cloud completion. Since the training data $X$ and $Y$ are not given as pairs, there may be a ***class imbalance problem***. For instance, consider point cloud data consisting of 'Chair' and 'Table' classes. The ratio of these two classes may differ between the incomplete point cloud distribution $\mu$ and the complete point cloud distribution $\nu$. While the incomplete point cloud data might consist of 50% 'Chair' and 50% 'Table,' the complete point cloud data could be composed of 70% 'Chair' and 30% 'Table.'

Unfortunately, the standard optimal transport problem (Eq. 1) is susceptible to this class imbalance problem (Eyring et al., 2024). The standard optimal transport map transports each source sample $x \sim \mu$ to a target sample $y \sim \nu$ without any rescaling. Consequently, in this class imbalance case, 20% of the 'Table' incomplete point cloud data would be transported to 20% of the 'Chair' complete point cloud. This behavior is undesirable for a point cloud completion model. In practice, **this**

---

**Algorithm 1** Training algorithm of UOT-UPC

---

**Require:** The mixture of the incomplete and complete point cloud distribution $\mu$. The complete
    point cloud distribution $\nu$. $\Psi_i^*(x) = \text{Softplus}(x)$. Generator network $T_\theta$ and the discriminator
    network $v_\phi$. $dl$ is density loss. Total iteration number $K$.
1: **for** $k = 0, 1, 2, \ldots, K$ **do**
2:     Sample a batch $X \sim \mu$, $Y \sim \nu$.
3:     $\mathcal{L}_{v,T} = \frac{1}{|X|} \sum_{x \in X} \Psi_1^* \left( -c\left(x, T_\theta(x)\right) + v_\phi\left(T_\theta(x)\right) \right) + \frac{1}{|Y|} \sum_{y \in Y} \Psi_2^*(-v_\phi(y)) - dl\left(T_\theta(x)\right)$
4:     Update $\theta$ by maximizing the loss $\mathcal{L}_{v,T}$.
5:     Update $\phi$ by minimizing the loss $\mathcal{L}_{v,T}$.
6: **end for**

---

**class imbalance problem occurs in the unpaired point cloud completion benchmark (Table 2).** In the multi-category case, the proportion of some categories, e.g., 'lamp' and 'trash bin' classes, significantly differs by more than threefold between the incomplete and complete point cloud distributions. To address this issue, we suggest the unbalanced optimal transport map as our point cloud completion model. **The robustness of UOT to class imbalance will be explained in Sec 3.2 and empirically demonstrated through experiments in Sec 5.2.**

### 3.2 ESTIMATION OF UNBALANCED OPTIMAL TRANSPORT MAP

In this section, we propose our point cloud completion model, which is based on the unbalanced optimal transport map, called UOT-UPC. Our goal is to learn the unbalanced optimal transport map $T^\star$ from the incomplete point cloud distribution $\mu$ to the complete point cloud distribution $\nu$ using a neural network $T_\theta$. To this end, we adopt the UOTM framework (Choi et al., 2023). This approach is based on the following semi-dual formulation of the UOT problem (Eq. 4, Vacher & Vialard (2023)).

$$C_{uot}(\mu, \nu) = \sup_{v \in \mathcal{C}} \left[ \int_{\mathcal{X}} -\Psi_1^* \left( -v^c(x) \right) d\mu(x) + \int_{\mathcal{Y}} -\Psi_2^*(-v(y)) d\nu(y) \right], \tag{6}$$

where the $c$-transform of $v$ is defined as $v^c(x) = \inf_{y \in \mathcal{Y}} (c(x, y) - v(y))$. We refer to the optimal maximizer $v^\star$ of Eq. 6 as the optimal potential function for the UOT problem. Following previous approaches for learning the optimal maps (Korotin et al., 2021; Fan et al., 2022; Rout et al., 2022; Choi et al., 2023), we introduce $T_\theta$ to approximate the unbalanced optimal transport map $T^\star$ as follows:

$$T_\theta(x) \in \operatorname*{arginf}_{y \in \mathcal{Y}} \left[ c(x, y) - v(y) \right] \quad \Leftrightarrow \quad v^c(x) = c\left(x, T_\theta(x)\right) - v\left(T_\theta(x)\right), \tag{7}$$

Note that the unbalanced optimal transport map $T^*$ satisfies the above conditions (Eq. 7) with the optimal potential $v^\star$ (Choi et al., 2023). By parametrizing the optimal potential $v^\star$ with a neural network $v_\phi$ and substituting $v^c$ using the right-hand side of Eq. 6, we arrive at the following learning objective $\mathcal{L}_{v_\phi, T_\theta}$:

$$\mathcal{L}_{v_\phi, T_\theta} = \inf_{v_\phi} \left[ \int_{\mathcal{X}} \Psi_1^* \left( -\inf_{T_\theta} \left[ c\left(x, T_\theta(x)\right) - v_\phi\left(T_\theta(x)\right) \right] \right) d\mu(x) + \int_{\mathcal{Y}} \Psi_2^* \left(-v_\phi(y)\right) d\nu(y) \right]. \tag{8}$$

Note that the learning objective $\mathcal{L}_{v_\phi, T_\theta}$ becomes the standard optimal transport map when the generator functions of $f$-divegence $\Psi_i$ are the convex indicator function at $\{1\}$, which means that its convex conjugate $\Psi_i^*$ is the identity function. Moreover, when the optimal potential $v^\star$ is given, the unbalanced optimal transport map can be interpreted as the optimal transport map between $\pi_0(x) = \Psi_1^{*\prime}(-v^{\star c}(x))\mu(x)$ and $\pi_1(y) = \Psi_2^{*\prime}(-v^\star(y))\nu(y)$ (Choi et al., 2023). **These rescaling factors $\Psi_i^{*\prime}(\cdot)$ offer the flexibility of the UOT map to handle the class imbalance problem (Eyring et al., 2024).** Our main contribution lies in formulating unpaired point cloud completion as the optimal transport problem, investigating the optimal cost function for this task, and applying this cost function within the UOTM framework.

### 3.3 IMPLEMENTATION DETAIL

As described in Algorithm 1, $\mathcal{L}_{v_\phi, T_\theta}$ can be computed by the Monte Carlo approximation with mini-batch samples from the incomplete point cloud $x$ and the complete point cloud $y$. Intuitively,

our learning objective is similar to the adversarial training in GANs (Goodfellow et al., 2020). Our potential $v_\phi$ and completion model $T_\theta$ play similar roles as the discriminator and generator in GANs, respectively. This is because the minimization with respect to $T_\theta$ in Eq 8 is equivalent to the maximization of $\mathcal{L}_{v_\phi, T_\theta}$[1].

We parametrize the generator and discriminator using the similar backbone network as USSPA (Ma et al., 2023) (See Appendix A for the implementation details). The InfoCD cost function $\text{InfoCD}(\cdot, \cdot)$ (Lin et al., 2024) is adopted as the cost function $c(\cdot, \cdot)$ in the learning objective $\mathcal{L}_{v_\phi, T_\theta}$. Moreover, in practice, we set the source distribution $\tilde{\mu}$ as a mixture of the incomplete point cloud distribution $\mu$ and complete point cloud distribution $\nu$, with a mixing probability of $50\%$, i.e., $\tilde{\mu} = 0.5\mu + 0.5\nu$. Then, we train the unbalanced optimal transport map between $\tilde{\mu}$ and $\nu$. This mixture trick helps our generator to produce high-fidelity complete point clouds. **We conducted ablation studies on the mixture trick and the cost function in Sec 5.3.**

## 4 RELATED WORKS

**Unpaired Point Completion Model** Unpaired point completion models have developed following recent advancements in unsupervised learning. Unpaired (Chen et al., 2020) is one of the first approaches for unpaired point completion. This model introduces a GAN-based model that maps the latent features of the incomplete point cloud to the latent features of the complete point cloud. Wu et al. (2020) proposes a conditional GAN model that generates multiple plausible complete point clouds conditioned on the incomplete point cloud. ShapeInv (Zhang et al., 2021) employs an optimization-based GAN-inversion approach (Xia et al., 2022). ShapeInv finds the optimal generator input noise to reconstruct the complete point cloud from the given incomplete point cloud. This is conducted by minimizing the distance between the input incomplete point cloud, which is for completion, and the partial point cloud, which is obtained by degrading the generator's output. Cycle4 (Wen et al., 2021) proposes two simultaneous cyclic transformations between the latent spaces of incomplete point cloud and complete one through missing region coding. USSPA (Ma et al., 2023) proposes a symmetric shape-preserving method based on GAN. This method utilizes a two-part generator. The first part is a coarse predictor with a symmetry learning module. The second part is an autoencoder with local feature grouping and an upsampling module. In this paper, we propose an unbalanced optimal transport approach for point cloud completion. To the best of our knowledge, this is the first attempt to introduce the optimal transport map for the unpaired point cloud completion.

## 5 EXPERIMENTS

In this section, we evaluate our model from various perspectives. For implementation details of experiments, please refer to Appendix A.

- In Sec 5.1, we evaluate our model on the unpaired point cloud completion benchmark, considering both single-category and multi-category settings.
- In Sec 5.2, we demonstrate the advantages of the UOT framework over the standard OT approach and the other point cloud completion model by testing under the class imbalance problem.
- In Sec 5.3, we conduct various ablations studies to investigate the effects of different cost functions, the source mixture trick, and the cost-intensity hyperparameters $\tau$.

### 5.1 UNPAIRED POINT COMPLETION PERFORMANCE

**Experimental Settings** In this section, we present both qualitative and quantitative results for unpaired point cloud completion using our model. We train and evaluate our model on the dataset proposed in Ma et al. (2023), which comprises ten categories, including chairs, trash bins, lamps, etc. To ensure a reliable and comprehensive comparison, we evaluate our model on (i) individual categories (*Single-category*) and (ii) all categories combined (*Multi-category*). In the single-category experiments, each model is trained and evaluated exclusively on data from a single class. In contrast,

---

[1]Since we assume $\Psi_i$ to be convex and non-negative, its convex conjugate $\Psi_i^*$ is an increasing function.

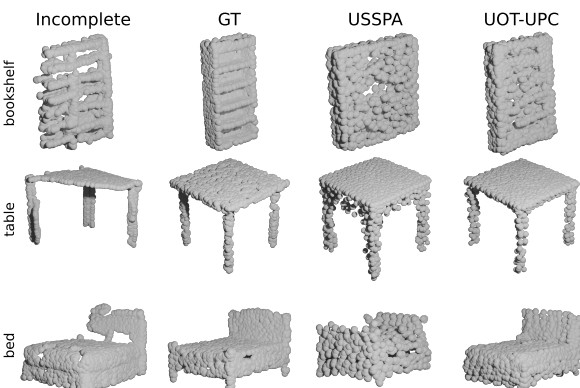

Figure 2: **Comparison of generated samples** from UOT-UPC and USSPA in the single-category.

Table 3: **Point cloud completion comparison in the single-category setting**, assessed by L1 Chamfer Distance $cd^{l1} \times 10^2$ ($\downarrow$). The boldface denotes the best performance among unpaired methods.

|  | Method | AVG | chair | table | trash bin | TV | cabinet | bookshelf | sofa | lamp | bed | tub |
|---|---|---|---|---|---|---|---|---|---|---|---|---|
| Paired | PoinTr (Yu et al., 2021) | 14.37 | 13.65 | 12.52 | 15.26 | 12.69 | 17.32 | 13.99 | 12.36 | 17.05 | 15.13 | 13.77 |
|  | Disp3D (Wang et al., 2022) | 7.78 | 6.24 | 8.20 | 7.12 | 7.12 | 10.36 | 6.94 | 5.60 | 14.03 | 6.90 | 5.32 |
|  | TopNet (Tchapmi et al., 2019) | 7.07 | 6.39 | 5.79 | 7.40 | 6.26 | 8.37 | 7.02 | 5.94 | 8.50 | 7.81 | 7.25 |
| Unpaired | ShapeInv (Zhang et al., 2021) | 21.39 | 17.97 | 17.28 | 33.51 | 15.69 | 26.26 | 25.51 | 14.28 | 16.69 | 32.33 | 14.43 |
|  | Unpaired (Chen et al., 2020) | 10.47 | 8.41 | 7.52 | 12.08 | 6.72 | 17.45 | 9.95 | 6.92 | 19.36 | 10.04 | 6.22 |
|  | Cycle4 (Wen et al., 2021) | 11.53 | 9.11 | 11.35 | 11.93 | 8.40 | 15.47 | 12.51 | 10.63 | 12.25 | 15.73 | 7.92 |
|  | USSPA (Ma et al., 2023) | 8.56 | 8.22 | 7.68 | 10.36 | 7.66 | **10.77** | 7.84 | **6.14** | 11.93 | **8.20** | 6.75 |
|  | UOT-UPC (Ours) | **7.62** | **7.88** | **6.44** | **8.83** | **6.00** | 11.84 | **7.32** | 6.65 | **7.30** | 8.69 | **5.49** |

the multi-category experiments use data from all classes for both training and evaluation. The multi-category setting is particularly challenging, as the model should learn to complete partial point clouds from diverse categories. For quantitative evaluation, we utilize the L1 Chamfer distance (Fan et al., 2017) ($cd^{l1}$) and F-scores (Tatarchenko et al., 2019) ($F_{score}^{0.1\%}$, $F_{score}^{1\%}$). These scores evaluate our completion results against the ground-truth completion on the test data. Further details on training procedures and evaluation metrics are provided in Appendix A.

**Single-category** In the single-category setting, we compare our model against existing point cloud completion models, including paired (supervised) and unpaired models. Fig. 2 illustrates the generated samples and Table 3 presents the L1 Chamfer distance ($cd^{l1}$) results (See Appendix B.2 for generated samples in the multi-category and Table 9 in the Appendix for results on the PCN dataset). Our model outperforms other unpaired models in seven out of ten categories in terms of $cd^{l1}$. The average column (AVG) indicates the average $cd^{l1}$ scores across all ten categories. In the AVG column, our model surpasses the second-best unpaired approach, USSPA (Ma et al., 2023), by more than 10% and even outperforms two paired approaches, PoinTr (Yu et al., 2021) and Disp3D (Wang et al., 2022). In particular, our model outperforms all other models, including the supervised ones, on TV and lamp datasets. Moreover, Table 4 reports the average of F-scores across all ten categories, following the evaluation scheme of Ma et al. (2023). Our model attains $F_{score}^{0.1\%}$ and $F_{score}^{1\%}$ scores of 19.55 and 76.83, respectively, surpassing all other unpaired methods. To sum up, our model consistently outperforms other unpaired point cloud models on most of the single-category datasets.

**Multi-category** Table 4 presents the $cd^{l1}$ and F-scores in the multi-category setting. Note that since this setting considers the entire dataset at once, the reported scores can be understood as a weighted sum of scores for each category, where the weights correspond to the ratio of training data in Table 2. Our model achieves $F^{0.1\%}$ score of 17.84, outperforming all other unsupervised benchmarks. Additionally, our model attains $cd^{l1} = 8.96$ and $F_{score}^{1\%} = 71.23$, which are comparable to the best-performing unpaired model, USSPA. In summary, our model shows comparable or better performance than the state-of-the-art model in multi-category point cloud completion.

Table 4: **Point cloud completion comparison** in the single-category setting and the multi-category setting, assessed by L1 Chamfer Distance $cd^{l1} \times 10^2$ ($\downarrow$) and F-scores $F_{score}^{0.1\%} \times 10^2$, $F_{score}^{1\%} \times 10^2$ ($\uparrow$).

| | Method | Single-category | | Multi-category | | |
| | | $F_{score}^{0.1\%} \uparrow$ | $F_{score}^{1\%} \uparrow$ | $cd^{l1} \downarrow$ | $F_{score}^{0.1\%} \uparrow$ | $F_{score}^{1\%} \uparrow$ |
|---|---|---|---|---|---|---|
| Paired | PoinTr (Yu et al., 2021) | - | - | 14.37 | 18.35 | 80.41 |
| | Disp3D (Wang et al., 2022) | - | - | 7.78 | 30.29 | 78.26 |
| | TopNet (Tchapmi et al., 2019) | - | - | 7.07 | 12.33 | 80.37 |
| Unpaired | ShapeInv (Zhang et al., 2021) | 15.58 | 66.53 | 19.35 | 16.98 | 69.66 |
| | Unpaired (Chen et al., 2020) | 12.20 | 64.33 | 10.12 | 10.86 | 66.68 |
| | Cycle4 (Wen et al., 2021) | 9.98 | 60.14 | 12.00 | 8.61 | 56.57 |
| | USSPA (Ma et al., 2023) | 17.49 | 73.41 | **8.96** | 16.88 | **72.31** |
| | UOT-UPC (Ours) | **19.55** | **76.83** | **8.96** | 17.84 | 71.23 |

## 5.2 Robustness to Class Imbalance of UOT approach

In this section, we explore the robustness of our model in class-imbalanced settings. As described in Sec 3.2, a key advantage of the UOT framework is its robustness and stability in handling *class imbalance* scenarios (Eyring et al., 2024). When the proportions of data classes between the source and target distributions differ, UOT can rescale the mass to compensate for this imbalance, ensuring that the learned transport map remains meaningful and accurate. Furthermore, note that this class imbalance is neither an unusual nor a contrived scenario. As we observed in Table 2, this class imbalance exists in even our multi-category experiment in Sec. 5.1.

**Experimental Settings** To explore the effects of class imbalance, we observe how the performance of existing point cloud completion models changes with different class imbalance ratios. To be more specific, we select two categories of datasets: Data1 (category: TV) and Data2 (category: Table). These categories are selected because of their relatively abundant training samples and the distinct differences in their shape. For the incomplete point cloud samples, we use the entire training data for both Data 1 and Data2, maintaining their ratio of $6.4 : 21.3$ in Table 2. For the complete point cloud samples, **we manipulate the imbalance ratio** $r$, i.e., Data1 and Data2 are sampled at a ratio of $6.4 : 21.3 \times r$. Then, each model is evaluated across diverse values of $r$ to explore the effects of class imbalance. We compare our model to (i) the standard OT counterpart of our model (OT-UPC) and (ii) USSPA, the state-of-the-art method for unpaired point cloud completion. Note that, as discussed in Sec. 3.2, our model corresponds to the standard OT counterpart when $\Psi_i^* = Id$. For detailed hyperparameter settings, please refer to Appendix A.

**Discussion** As shown in Table 5, our model outperforms the two alternative models across various class imbalance settings. (See Table 8 in the Appendix for results on other class combinations.) Note that we tested $r \leq 1$, because Data2 has a significantly larger total number of training samples, more than three times that of Data1 (Table 2). Hence, setting $r > 1$ would result in discarding too many training data samples. Our model consistently demonstrates stable performance, ranging between 6.65 and 6.78 across various class imbalance ratios $r$, while USSPA shows considerably greater variance. In contrast, the standard OT generally performs poorly, with its best result appearing in the balanced case (1:1 ratio). We hypothesize that this phenomenon occurs due to the unstable training dynamics of the standard OT. The stable training dynamics in learning the transport map is also another advantage of the UOT over OT (Choi et al., 2024). In summary, these results indicate that our UOT-UPC offers strong robustness to class imbalance problem.

## 5.3 Ablation Study

**Effect of Appropriate Cost Functional** We validate our motivation experiments (Table 1) for selecting InfoCD (Lin et al., 2024) as the cost function. In the (unbalanced) optimal transport map approach, the cost function $c(\cdot, \cdot)$ in Eq. 8 determines how each input $x$ is transported to the $y = T^*(x)$ by the optimal transport map $T^*$. Thus, setting an appropriate cost function is crucial. In this regard, as a reminder, we assessed various cost function options to determine whether their cost-minimizing pairs are suitable for the point cloud completion in Sec 3.1. Here, we conduct an ablation study by modifying the cost function $c(\cdot, \cdot)$ in our model (Eq. 8). Each model is evaluated on the multi-category setting and the single-category settings for the 'trash bin' and 'TV' classes.

Table 5: **Comparison of class imbalance robustness** ($cd^{l1} \times 10^2$ ($\downarrow$)) on (Data1, Data2) = (TV, Table).

| $r$ | 0.3 | 0.5 | 0.7 | 1 |
|---|---|---|---|---|
| USSPA | 7.60 | 6.97 | 8.08 | 7.97 |
| OT-UPC | 25.12 | 25.72 | 24.30 | 21.49 |
| Ours | **6.71** | **6.65** | **6.70** | **6.78** |

Table 6: **Ablation study on the cost function** $c(\cdot, \cdot)$ ($cd^{l1} \times 10^2$ ($\downarrow$)).

| Cost function | Multi-category | trash bin | TV |
|---|---|---|---|
| $l_2$ | 23.80 | 39.22 | 19.19 |
| $cd^{l2}$ | 10.05 | 10.57 | 6.37 |
| $cd^{l2}_{fwd}$ | 13.19 | 10.05 | 7.23 |
| InfoCD | **8.96** | **8.83** | **6.00** |

Table 7: **Ablation study on the source mixture trick**, i.e., the complete input.

| Category | Complete Input | $cd^{l1} \downarrow$ | $F_{\text{score}}^{0.1\%} \uparrow$ | $F_{\text{score}}^{1\%} \uparrow$ |
|---|---|---|---|---|
| Single | | 7.90 | 17.40 | 74.11 |
| | ✓ | **7.62** | **19.55** | **76.83** |
| Multi | | 9.00 | 16.66 | 70.86 |
| | ✓ | **8.96** | **17.84** | **71.23** |

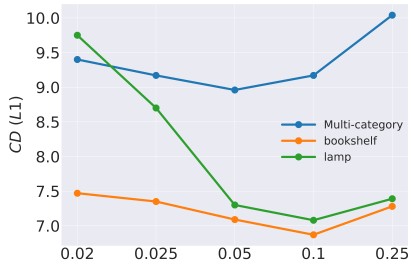

Figure 3: **Ablation study on the cost intensity** $\tau$ ($cd^{l1} \times 10^2$ ($\downarrow$)).

Table 6 demonstrates that our model achieves the best performance using the InfoCD cost function, followed by ($cd^2_{fwd}$, $cd^2$), and $l^2$. (See Table 10 for the cost ablation results on the PCN dataset.) Note that this ranking closely aligns with the results of our cost function investigation in Table 1. This consistency suggests a strong correlation between our motivation experiments and actual model performance. Furthermore, these findings suggest that further exploration of alternative cost functions could potentially enhance our model's performance. We leave this exploration for future work.

**Add Complete Sample to Source** As described in Sec 3.3, we introduced the source mixture trick to our model, i.e., the source distribution is given as a mixture of incomplete and complete point cloud data with a mixing probability of 50%. Here, we conduct an ablation study to evaluate the effect of this source mixture trick. The results are presented in Table 7. In both single-category and multi-category experiments, our model exhibits consistent improvements in both $cd^{l1}$ and F scores with the source mixture trick. The purpose of this source mixture trick is to assist our transport map in generating the target distribution better. For input complete data, the optimal transport map should ideally learn the identity mapping, which is relatively easier compared to completing the input incomplete point cloud. We hypothesize this property encourages the training process, enabling the model to generate complete point clouds more efficiently. Therefore, we empirically observed an improvement in the fidelity of the point cloud completion when using this source mixture trick.

$\tau$ **Robustness** For the last ablation study, we evaluate the robustness of our model with respect to the cost-intensity hyperparameter $\tau$, defined as $c(x, y) = \tau \times \text{InfoCD}(x, y)$. Specifically, we tested our model on the multi-category setting and the single-category settings of the 'bookshelf' and 'lamp' classes, while changing $\tau \in \{0.02, 0.025, 0.05, 0.1, 0.25\}$. Note that we impose challenging conditions by setting the maximum $\tau$ to $\tau_{\max} = 0.25$ and the minimum $\tau$ to $\tau_{\min} = 0.02$, resulting in a ratio of $\tau_{\max}/\tau_{\min} > 10$. As depicted in Fig. 3, our model shows moderate performance across various $\tau$ values. In particular, the sweet spot of $\tau$ lies roughly between 0.05 and 0.1. The performance deteriorates by approximately 10% when $\tau$ is either too large ($\tau_{\max}$) or too small ($\tau_{\min}$).

## 6 CONCLUSION

In this paper, we introduce UOT-UPC, an unpaired point cloud completion model based on the UOT map. To the best of our knowledge, our work is the first attempt to introduce the unbalanced optimal transport map to the point cloud completion task. We formulated the unpaired point cloud completion task as an (unbalanced) optimal transport problem and investigated the optimal cost function for this task. Our experiments demonstrated a strong correlation between cost function selection and the model's point cloud completion performance. When combined with the InfoCD cost function, our UOT-UPC attains competitive performance compared to both unpaired and paired point cloud completion models. Moreover, our experiments showed that UOT-UPC presents robustness to the class imbalance problem, which is prevalent in the unpaired point cloud completion tasks.

ETHICS STATEMENT

The point cloud completion research contributes positively to various fields, including autonomous driving, robotics and virtual/augmented reality. Also, it is applicable to urban planning and cultural heritage preservation. Our research does not involve personal data or human subjects, and we have carefully addressed potential data bias issues. We also ensure that there are no risks related to illegal surveillance or privacy violations. As a result, we believe that this research is conducted ethically and poses no social or ethical concerns.

REPRODUCIBILITY STATEMENT

To ensure the reproducibility of our work, we submitted the anonymized source in the supplementary material and included the implementation and experiment details in Appendix A.

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

## A    Implementation Details

Unless otherwise stated, our implementation follows the experimental settings and hyperparameters of USSPA Ma et al. (2023).

### A.1    Network

We adopt the generator and discriminator architectures from the USSPA framework as completion model $T_\theta$ and potential $v_\phi$. For the potential $v_\phi$, the final sigmoid layer of the discriminator is omitted to allow for the parameterization of the potential function, enabling outputs to assume any real values. Additionally, we remove the feature discriminator to streamline the architecture. In the potential $v_\phi$, we implement the encoder proposed by Yuan et al. (2018) in their Point Cloud Networks (PCN). Following the encoder, we employ an MLPConv layer specified as $\text{MLPConv}(C_{in}, [C_1, \ldots, C_n]) = \text{MLPConv}(1024, [256, 256, 128, 128, 1])$, which indicates that the output $y$ is computed as follows:

$$y = \text{Conv1D}_{C_4=128, C_5=1}(\text{ReLU}(\ldots \text{ReLU}(\text{Conv1D}_{C_{in}=1024, C_1=256}(x))\ldots)) \tag{9}$$

Here, $\text{Conv1D}_{C_{\text{in}}, C_{\text{out}}}$ represents a 1D convolutional layer with $C_{\text{in}}$ input channels and $C_{\text{out}}$ output channels.

The completion model $T_\theta$ receives as input a concatenation of the incomplete point cloud and a complete point cloud. These inputs are processed independently to generate distinct complete point cloud samples. The completion model $T_\theta$ follows an Encoder-Decoder architecture, augmented by an upsampling refinement module (upsampling module) in sequence. The upsampling module is implemented using a 4-layer MLPConv network, where the final MLPConv layer is responsible for refining and adding detailed structures to the output (Ma et al., 2023). Specifically, the inputs to the last MLPConv layer are composed of the skeleton point cloud produced by the Encoder-Decoder structure and the features extracted from the third MLPConv layer.

### A.2    Implementation detail

**Motivation - Optimal Cost Function**    The incomplete and complete point clouds utilized in the optimal cost function outlined in Sec 3.1 are sourced from the dataset proposed by Ma et al. (2023). This dataset consists of paired incomplete and complete point clouds. For a fair comparison, we shuffle the complete point clouds to create an unpaired setting. We then use these shuffled point clouds as artificial complete data to train the USSPA model.

**Training**    Concerning the loss function $L_{v,T}$. We employ Infocd as the cost function $c$ with a coordinate value of $\tau = 0.05$. For the hyperparameters of InfoCD, we set $\tau_{\text{infocd}}$ to 2 and $\lambda_{\text{InfoCD}}$ to $1.0 \times 10^{-7}$. The functions $\Psi_1^*$ and $\Psi_2^*$ are defined using the Softplus activation, $\text{SP}(x) = 2\log(1 + e^x) - 2\log 2$.[2] As a regularization term, we incorporate the density loss $dl$ proposed by Ma et al. (2023), and we designate a coordinate value of 10.5 for $dl$. The objective of Potential $v_\phi$ is to assign high value to target sample $y$ while assigning lower values to generated sample $\hat{y}$. We utilize the Adam optimizer with $\beta_1 = 0.95, \beta_2 = 0.999$ and learning rates of $2.0 \times 10^{-5}, 1.0 \times 10^{-5}$ for the potential $v_\phi$ and completion model $T_\theta$, respectively. The training is conducted with a batch size 4. The maximum epoch of training is 480. We report the final results based on the epoch that yields the best performance.

**Ablation study - Effect of Appropriate Cost Functional**    We set cost function coordinate value $\tau = 100$ for cost function $cd^{l2}{}_{fwd}, cd^{l2}$ and $l^2$. All other parameters and settings, unless otherwise specified, are consistent with those used in our UOT-UPC model.

---

[2]The softplus function is translated and scaled to satisfy $\text{SP}(0) = 0$ and $\text{SP}'(0) = 1$.

**Evaluation Metrics**

- $L1$-Chamfer Distance $cd^{l1}$ (Fan et al., 2017)

$$cd^{l1}(x_i, y_j) = \frac{1}{2}\left(\frac{1}{|x_i|}\sum_m \min_n \|x_{im} - y_{jn}\|_2 + \frac{1}{|y_j|}\sum_n \min_m \|x_{im} - y_{jn}\|_2.\right) \quad (10)$$

where each of $x_i, y_j$ is point cloud

- $F$ score $F_{score}^\alpha$ (Tatarchenko et al., 2019)

$$F_{score}^\alpha = \frac{2 \times P(\alpha) \times R(\alpha)}{P(\alpha) + R(\alpha)} \quad (11)$$

where $P(\alpha) = \frac{|\{x_{im} \in x_i \,|\, \min_n(\|x_{im} - y_{jn}\|_2) < \alpha\}|}{|x_i|}$ measures the accuracy of $x_i$,

and $R(\alpha) = \frac{|\{y_{jn} \in y_j \,|\, \min_m(\|x_{im} - y_{jn}\|_2) < \alpha\}|}{|y_j|}$ measures the completeness of $x_i$.

## A.3 OT-UPC

For the completion model $T_\theta$, we implement MLPConv$(512, [128, 128, 1])$ following the PCN encoder (Yuan et al., 2018). We incorporate R1 regularization (Roth et al., 2017) and R2 regularization (Mescheder et al., 2018) to the loss function $L_{v,T}$. Both regularization terms are assigned coordinate values $r1 = r2 = 0.2$. The density loss $dl$ is excluded from the $L_{v,T}$. A gradient clipping value of 1.0 is applied. We use Adam optimizer with $\beta_1 = 0.9, \beta_2 = 0.999$ and a learning rate $lr_{T_\theta} = 5.0 \times 10^{-5}$ for the completion model $T_\theta$. In addition, we use Adam optimizer with $\beta_1 = 0.9, \beta_2 = 0.999$ and learning rate $lr_{v_\phi} = 1.0 \times 10^{-7}$ for the potential $v_\phi$. All other settings not explicitly mentioned follow those of our model, UOT-UPC.

## B ADDITIONAL RESULTS

### B.1 ADDITIONAL VISUALIZATION OF THE THREE-NEAREST NEIGHBOR OF VARIOUS COST FUNCTIONS FROM SEC. 3.1

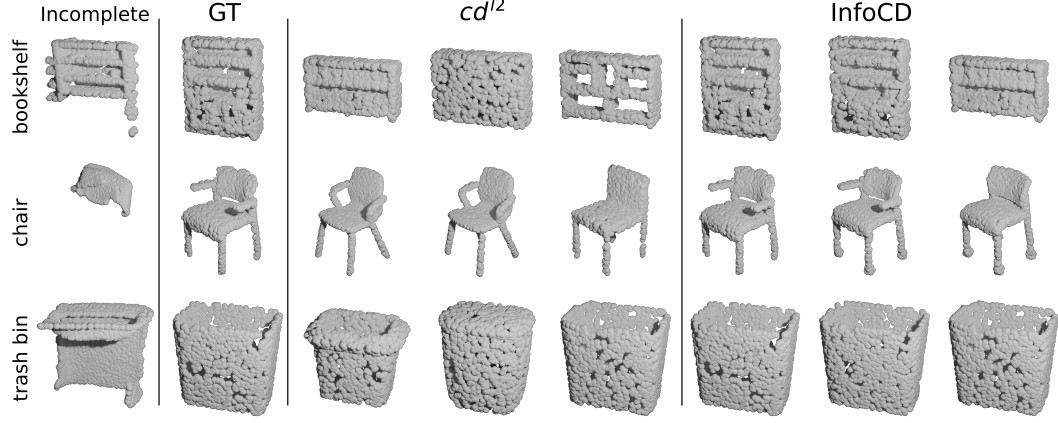

Figure 4: **Visualization of the incomplete point cloud** $x$**, the ground-truth completion** $y^{gt}(x)$**, and the three complete point clouds** $y_i^c(x)$ that minimize the cost $c(x, y_i^c(x))$ for two cost functions: $cd^{l2}$ and InfoCD, in the **single-category setting**.

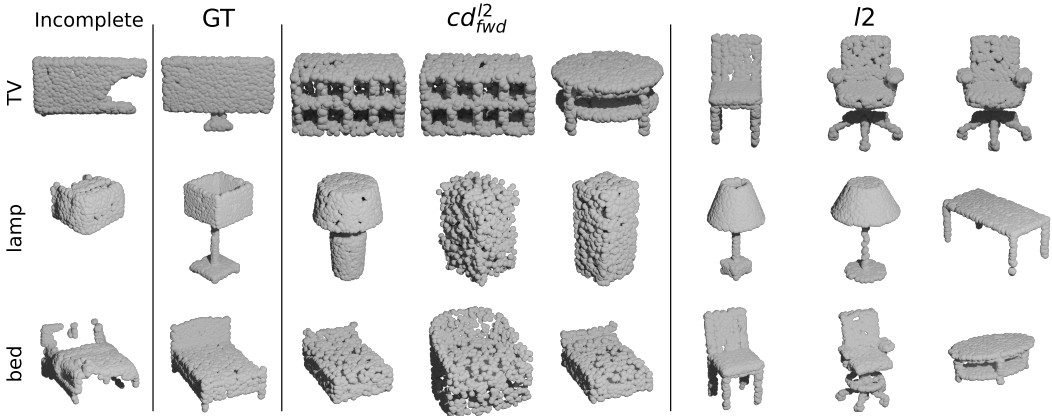

Figure 5: **Visualization of the incomplete point cloud** $x$**, the ground-truth completion** $y^{gt}(x)$**, and the three complete point clouds** $y_i^c(x)$ that minimize the cost $c(x, y_i^c(x))$ for two cost functions: $cd^{l2}{}_{fwd}$ and $l2$, in the **multi-category setting**.

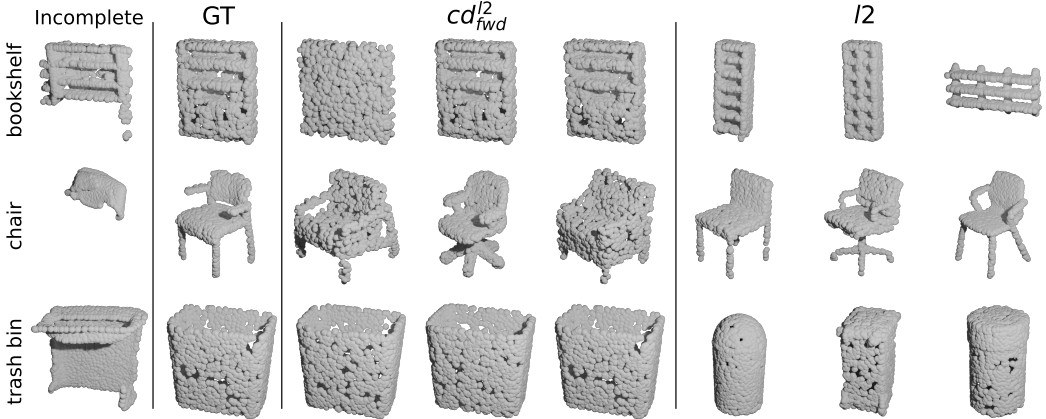

Figure 6: **Visualization of the incomplete point cloud** $x$**, the ground-truth completion** $y^{gt}(x)$**, and the three complete point clouds** $y_i^c(x)$ that minimize the cost $c(x, y_i^c(x))$ for two cost functions: $cd^{l2}{}_{fwd}$ and $l2$, in the **single-category setting**.

## B.2 ADDITIONAL QUALITATIVE RESULTS

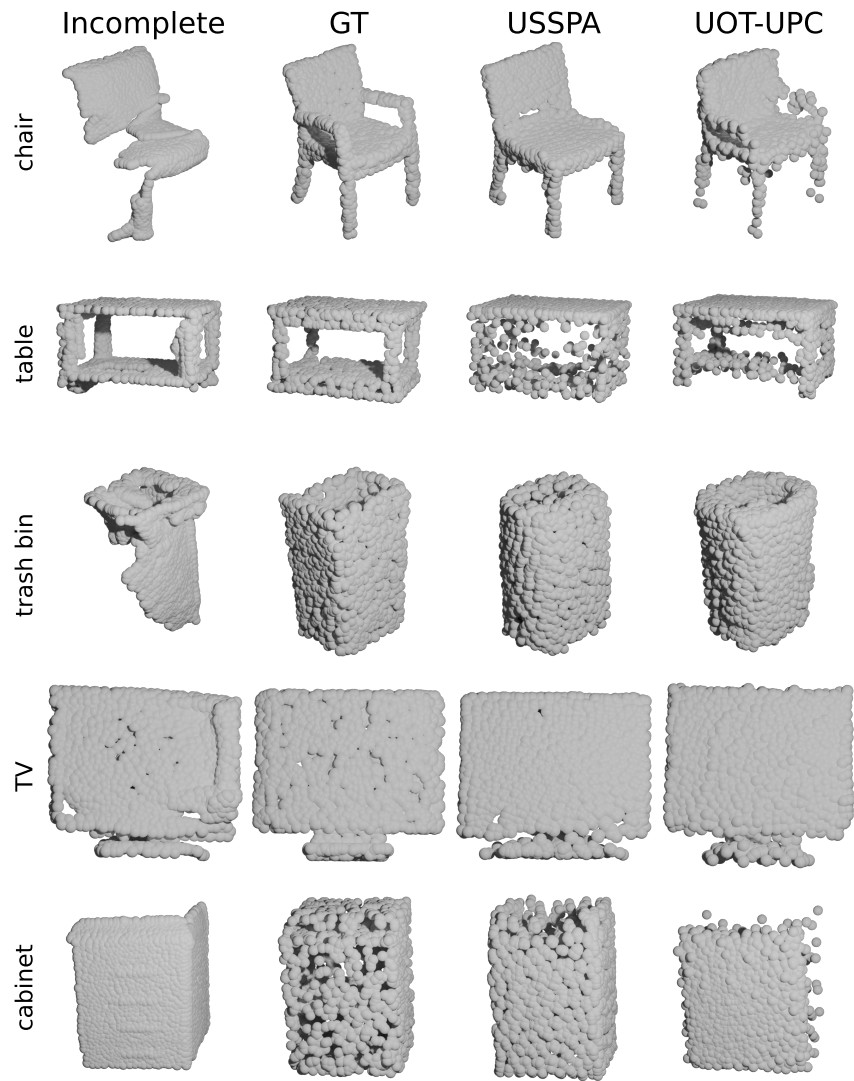

Figure 7: **Comparison of generated samples** from our UOT-UPC and USSPA in the single-category setting.

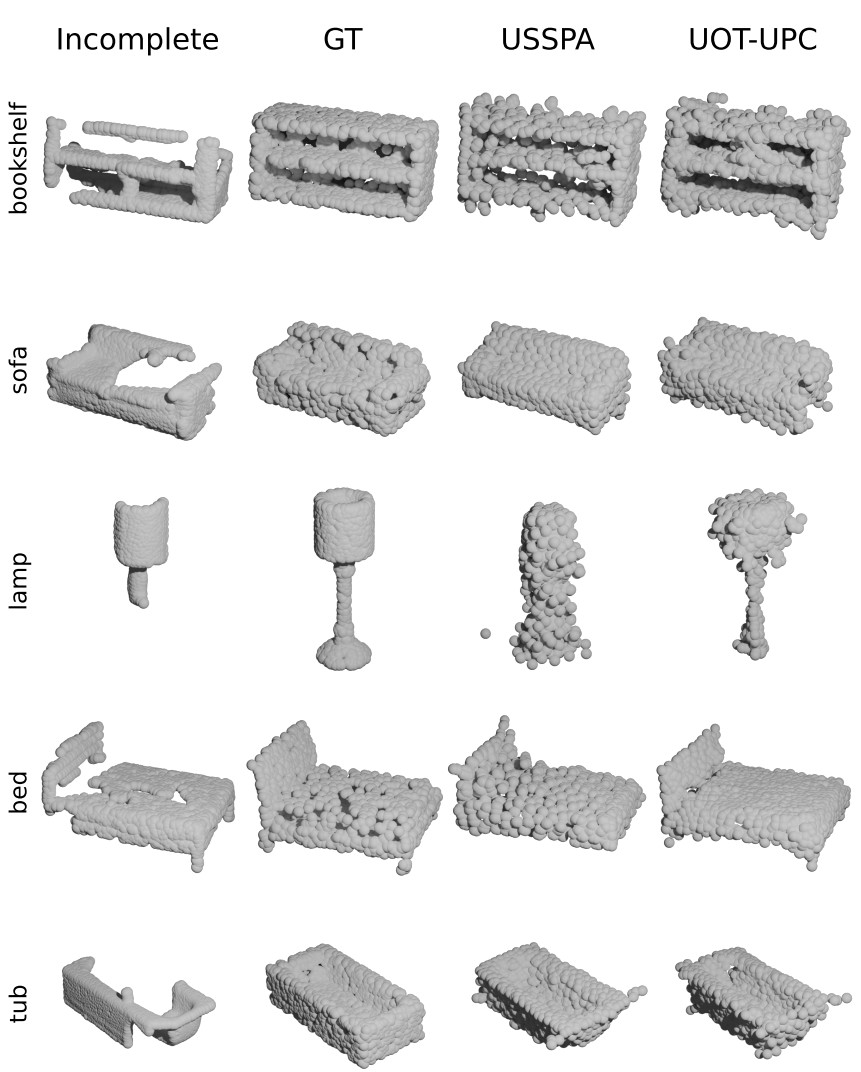

Figure 8: **Comparison of generated samples** from our UOT-UPC and USSPA in the single-category setting.

972
973
974
975
976
977
978
979
980
981
982
983
984
985
986
987
988
989
990
991
992
993
994
995
996
997
998
999
1000
1001
1002
1003
1004
1005
1006
1007
1008
1009
1010
1011
1012
1013
1014
1015
1016
1017
1018
1019
1020
1021
1022
1023
1024
1025

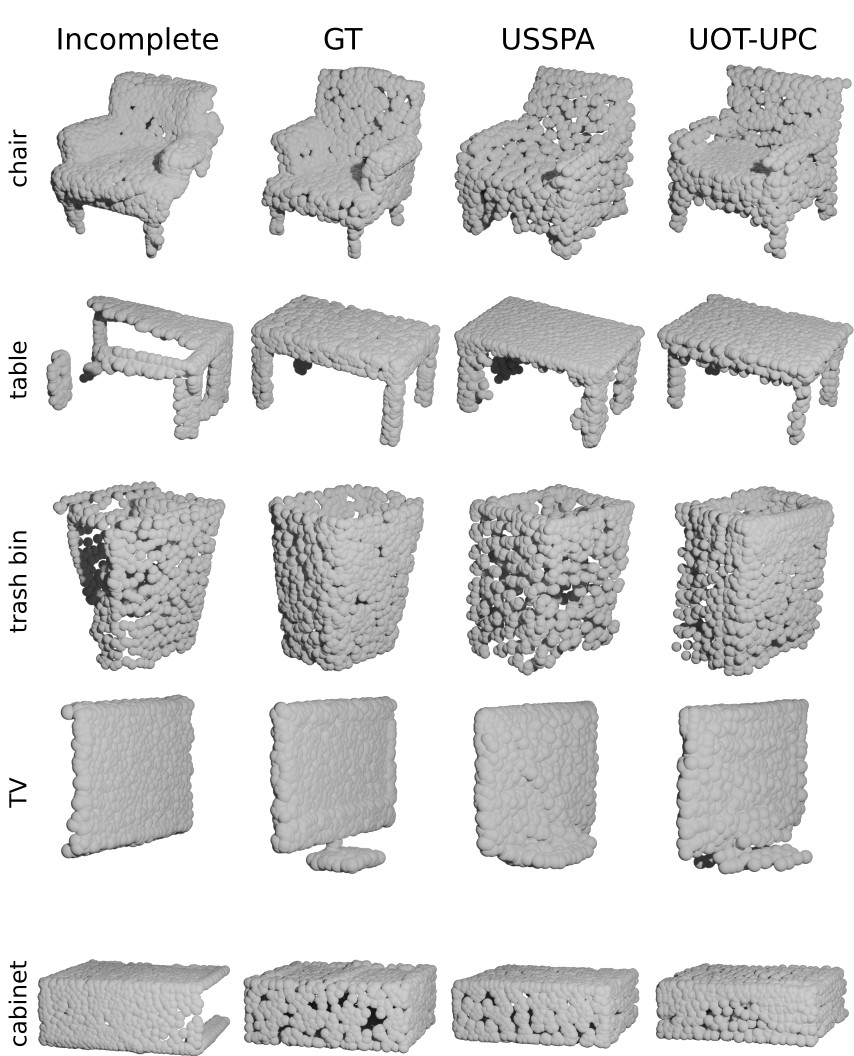

Figure 9: **Comparison of generated samples** from our UOT-UPC and USSPA in the multi-category setting.

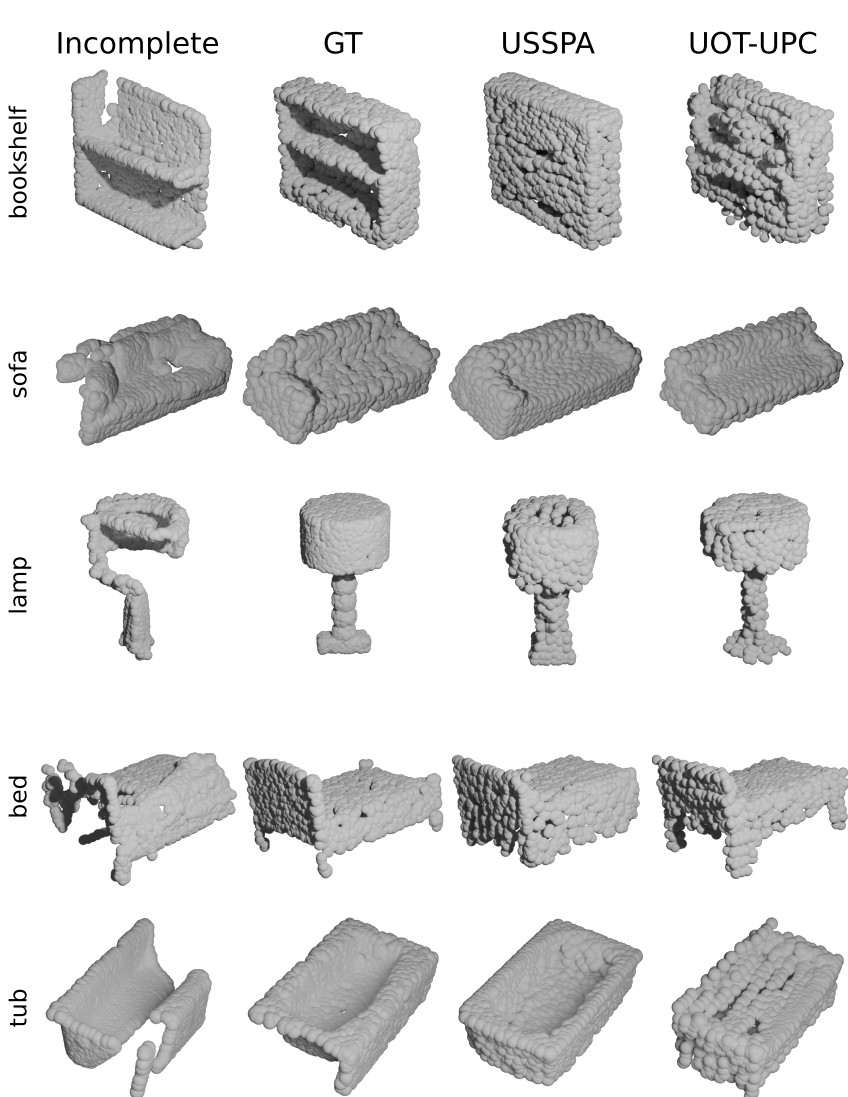

Figure 10: **Comparison of generated samples** from our UOT-UPC and USSPA in the multi-category setting.

### B.3 Qualitative comparison between our UOT-UPC and existing methods on the KITTI dataset.

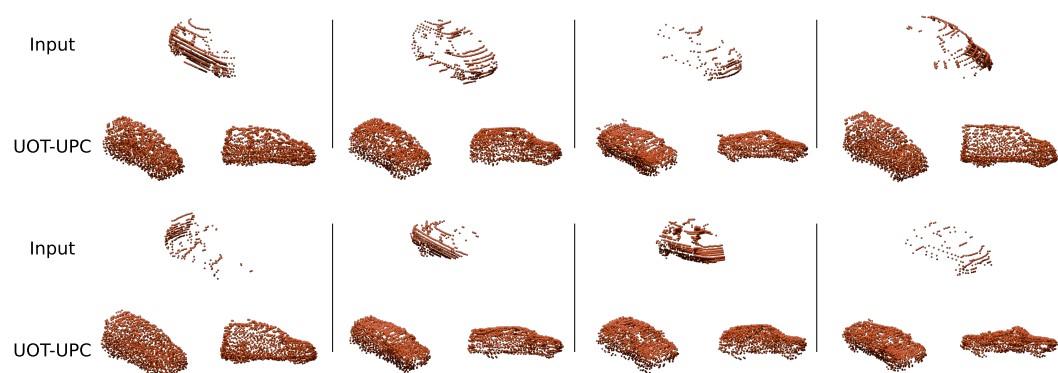

Figure 11: **Point cloud completion results of the UOT-UPC model on the KITTI dataset (Geiger et al., 2012)**. The model is trained on the ShapeNet dataset under the car category and tested on partial point clouds from the KITTI dataset without fine-tuning. From the qualitative comparison with previous approaches (Fig 12), our UOT-UPC model achieves higher-fidelity point cloud completion, demonstrating better global structure and more evenly distributed points.

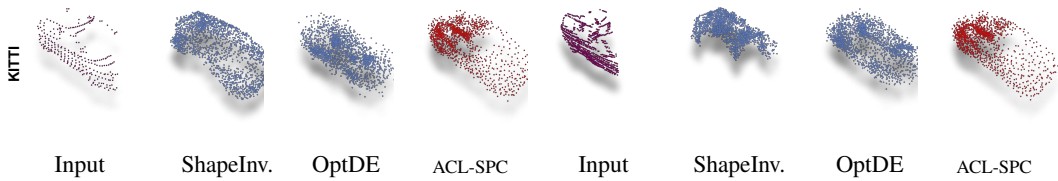

Figure 12: **Point cloud completion results of previous models on the KITTI dataset Geiger et al. (2012)**. The generated samples are taken from ACL-SPC (Hong et al., 2023), which is a self-supervised model. The others are unsupervised approaches: ShapeInv (Zhang et al., 2021) and OptDE (Gong et al., 2022).

## B.4 Comparison of class imbalance robustness for diverse class combinations.

Table 8: **Comparison of class imbalance robustness** ($cd^{l1} \times 10^2$ ($\downarrow$)) between UOT-UPC (ours), USSPA, and OT-UPC on diverse class combinations (Data1, Data2). Our UOT-UPC consistently outperforms other models across a wide range of class imbalance ratios in both additional class settings.

(a) (Data1, Data2) = (Lamp, Trash bin) with sample count = (1.1 : 8.0 * $r$).

| $r$ | 0.3 | 0.5 | 0.7 | 1 |
|---|---|---|---|---|
| USSPA | 10.16 | 9.49 | 10.21 | 10.21 |
| OT | 22.03 | 21.37 | 21.07 | 29.43 |
| Ours | **9.24** | **9.01** | **9.39** | **9.41** |

(b) (Data1, Data2) = (Lamp, Bed) with sample count = (1.1 : 2.9 * $r$).

| $r$ | 0.3 | 0.5 | 0.7 | 1 |
|---|---|---|---|---|
| USSPA | 9.64 | 9.78 | 9.27 | 9.79 |
| OT | 22.68 | 20.18 | 22.91 | 22.75 |
| Ours | **8.65** | **8.83** | **8.87** | **9.04** |

## B.5 Additional experimental results on the PCN dataset

Table 9: **Point cloud completion comparison** on the PCN dataset in the single-category setting, assessed by L1 Chamfer Distance $cd^{l1} \times 10^2$ ($\downarrow$). All unpaired models are trained with ScanNet. The boldface denotes the best performance among unpaired methods. Our UOT-UPC outperforms all other unpaired point cloud completion models.

| | Method | AVG | chair | table | cabinet | sofa | lamp |
|---|---|---|---|---|---|---|---|
| Paired | PoinTr (Yu et al., 2021) | 5.49 | 5.61 | 5.68 | 6.08 | 5.67 | 4.44 |
| | Disp3D (Wang et al., 2022) | 2.51 | 2.42 | 2.30 | 2.38 | 2.44 | 3.00 |
| | TopNet (Tchapmi et al., 2019) | 5.92 | 6.34 | 5.45 | 6.06 | 5.80 | 5.95 |
| Unpaired | ShapeInv (Zhang et al., 2021) | 19.05 | 23.18 | 15.66 | 17.14 | 22.85 | 16.40 |
| | Unpaired (Chen et al., 2020) | 14.87 | 12.87 | 8.14 | 14.30 | 18.23 | 20.82 |
| | Cycle4 (Wen et al., 2021) | 17.60 | 14.25 | 15.73 | 21.06 | 21.54 | 15.40 |
| | USSPA (Ma et al., 2023) | 12.63 | 13.52 | 9.66 | 8.89 | 15.51 | 15.57 |
| | UOT-UPC (Ours) | **7.92** | **10.22** | **8.11** | **6.41** | **8.08** | **6.79** |

Table 10: **Ablation study on the cost function** $c(\cdot, \cdot)$ on the PCN dataset ($cd^{l1} \times 10^2$ ($\downarrow$)). The results are consistent with Table 6. InfoCD achieved the best performance, while the L2 distance yielded the worst results.

| Cost function | cabinet | sofa | lamp |
|---|---|---|---|
| $l_2$ | 19.38 | 17.92 | 17.27 |
| $cd^{l2}$ | 8.52 | 8.33 | 7.23 |
| $cd^{l2}{}_{fwd}$ | 14.28 | 11.70 | 11.76 |
| InfoCD | **6.41** | **8.08** | **6.79** |

