# OpenReview forum: "Unsupervised Point Cloud Completion through Unbalanced Optimal Transport"
_ICLR.cc/2025/Conference — Submitted to ICLR 2025_

### Official Review · Reviewer_uLAt · 2024-10-29

**Soundness:** 2
**Presentation:** 2
**Contribution:** 2
**Rating:** 3
**Confidence:** 4

**Summary:**

This paper propose an unbalanced optimal transport for unpaired point cloud completion. They treat the completion problem as mapping incomplete set to complete set and utilize InfoCD for cost function. They show results for single-category and multiple-category unpaired point cloud completion and conducted many ablation study.

**Strengths:**

1. They propose to deal with the unbalance problem in point cloud completion.
2. They present better results than the competitors.

**Weaknesses:**

1. Optimal Transport Map is utilized to map the incomplete set to complete set. However, this assume the set of incomplete point cloud and the set of complete set should be complete set. For instance, this work utilized the complete point cloud from ShapeNet and partial point cloud from ScanNet. The original dataset only align the ShapeNet model with the real scans, however, the assumption in Optimal Transport is hard to be satisfied.
2. Experiments are only conducted in data from Scan2CAD. More experiments should be conducted on MatterPort3D or KITTI like "ACL-SPC: Adaptive Closed-Loop system for Self-Supervised Point Cloud Completion" and ModelNet or 3D-FUTURE like "CloudMix: Dual Mixup Consistency for Unpaired Point Cloud Completion".
3. More related works like "ACL-SPC: Adaptive Closed-Loop system for Self-Supervised Point Cloud Completion" and "CloudMix: Dual Mixup Consistency for Unpaired Point Cloud Completion" should be compared.

**Questions:**

Whether the assumption in optimal transport map is satisfied by the given data should be discussed.

---

> ### Author Response · Authors · 2024-11-21
> **Response to Reviewer uLAt(1/2)**
>
> We sincerely thank the reviewer for carefully reading our manuscript and providing valuable feedback. We hope our responses to be helpful in addressing the reviewer's concerns. We highlighted the corresponding revisions in the manuscript in Brown.
>
> $ $
>
> ---
>
> **W1, Q.** Optimal Transport Map is utilized to map the incomplete set to complete set. However, this assume the set of incomplete point cloud and the set of complete set should be complete set. For instance, this work utilized the complete point cloud from ShapeNet and partial point cloud from ScanNet. The original dataset only align the ShapeNet model with the real scans, however, the assumption in Optimal Transport is hard to be satisfied.
>
> **A.** We are not entirely certain about the specific "assumption" the reviewer is referring to. If our response does not address the reviewer’s concern, we kindly ask for clarification, and we would be happy to provide a follow-up response.
>
> First, if **the assumption is about the target data setting**, we evaluated our model on the dataset from USSPA [1]. As the reviewer commented, their benchmark is constructed by combining point clouds from two sources: partial point clouds from ScanNet (real scan) and complete point clouds from ShapeNet (synthetic). The motivation for this setup is that an unpaired approach is especially advantageous if we can train the completion model using the synthetic point cloud data, because it is infeasible to obtain the complete point cloud for real scans. In this respect, our results showed that the UOT map approach is effective for this setting. Specifically, learning the UOT map between the point cloud distributions from two sources provides meaningful completion. Moreover, we would like to emphasize that we demonstrated the appropriateness of this cost-minimizing pairing via the UOT map in Sec. 3.1 (Fig. 1 and Table 1).
>
>
> Furthermore, we would like to highlight that we conducted additional experiments on the PCN dataset, where **the complete and incomplete point clouds are from the same sources**. In this additional experiment, our model outperforms other baseline methods (Table 9).
> - Point cloud completion comparison on the PCN Dataset, assessed by L1 Chamfer Distance $cd^{l 1} \times 10^2$ ($\downarrow$).
>
> |   |Method | AVG | chair | table | cabinet | sofa | lamp |
> |:---|:---|:---|:---|:---|:---|:---|:---|
> |  Unpaired | ShapeInv | 19.05 | 23.18 | 15.66 | 17.14 | 22.85 | 16.40 |
> |  Unpaired | Unpaired | 14.87 | 12.87 | 8.14 | 14.30 | 18.23 | 20.82 |
> |Unpaired| Cycle4 | 17.60 | 14.25 | 15.73 | 21.06 | 21.54 | 15.40 |
> | Unpaired  | USSPA  | 12.63 | 13.52 | 9.66 | 8.89 | 15.51 | 15.57 |
> | Unpaired  | Ours | **7.92** | **10.22** | **8.11** | **6.41** | **8.08** | **6.79** |
>
>
>
> Second, if **the assumption is about the theoretical perspective**, the Optimal Transport Map is a general framework for connecting two probability distributions: the source and target distributions. The given cost function determines how each source sample and target sample are connected by this Optimal Transport Map (Eq. 1). Therefore, there are no specific assumptions on the source and target distributions, except for some technical assumptions in Lines 73 - 81.
>
>
> [1] Ma, Changfeng, et al. "Symmetric shape-preserving autoencoder for unsupervised real scene point cloud completion." CVPR 2023.
>
> $ $
>
> ---
> **W2** Experiments are only conducted in data from Scan2CAD. More experiments should be conducted on MatterPort3D or KITTI like "ACL-SPC: Adaptive Closed-Loop system for Self-Supervised Point Cloud Completion" and ModelNet or 3D-FUTURE like "CloudMix: Dual Mixup Consistency for Unpaired Point Cloud Completion".
>
> **A.** Thank you for suggesting the relevant references. Since we could not find the code implementation of CloudMix, **we conducted additional experiments on the KITTI dataset and compared our model with ACL-SPC and other unsupervised methods**. We included the qualitative completion samples in Fig 11 and 12 in the Appendix of our revised version manuscript. Fig 12 illustrates that our model exhibits better completions compared to other methods, particularly by providing more global completions.
>
> $ $
>
> ---

---

> > ### Author Response · Authors · 2024-11-21
> > **Response to Reviewer uLAt(2/2)**
> >
> > **W3.**
> > More related works like "ACL-SPC: Adaptive Closed-Loop system for Self-Supervised Point Cloud Completion" and "CloudMix: Dual Mixup Consistency for Unpaired Point Cloud Completion" should be compared.
> >
> > **A.** Thank you for suggesting SOTA methods for 3D shape completion. We additionally cited ACL-SPC and CloudMix in the first paragraph of the Introduction. We compared our model with ACL-SPC in the additional KITTI experiment in the response to W2. However, we respectfully believe that CloudMix is not directly comparable to our unpaired (unsupervised) model. CloudMix conducts point cloud completion via cross-domain adaptation. In contrast, our model focuses on direct unsupervised completion, without utilizing information from other domains. Hence, the target problems are different. Moreover, the training of CloudMix requires paired synthetic data, which distinguishes it further from our approach.

---

> > > ### Author Response · Authors · 2024-11-25
> > >
> > > We sincerely thank the reviewer for the effort in reviewing our paper. We would greatly appreciate it if the reviewer let us know whether our response was helpful in addressing the reviewer's concerns. If there are additional concerns or questions, please let us know. If our responses were helpful in addressing concerns, we kindly ask the reviewer to consider raising the score.

---

> > ### Comment · Reviewer_uLAt · 2024-11-26
> >
> > The results on PCN only present the performance of the proposed method when the complete shape and imcomplete shape are from the same source, which meet the assumption of this paper. However, in many cases such assumption cannot be satisfied. The results in Fig. 11 and Fig. 12 cannot compare the proposed method with other methods given the same inputs.

---

> > > ### Author Response · Authors · 2024-11-27
> > >
> > > ---
> > > **Q1.**
> > > The results on PCN only present the performance of the proposed method when the complete shape and imcomplete shape are from the same source, which meet the assumption of this paper. However, in many cases such assumption cannot be satisfied.
> > >
> > > **A.**
> > > Thank you for clarifying the "assumption" the reviewer is referring to. Indeed, **the distribution gap between the ground-truth complete point cloud distribution $\nu_{gt}$ and the training complete point cloud distribution from different source $\nu$ can be another reason for the success of the UOT approach.**
> > >
> > >  As clarified in our previous response, the OT framework is a general framework that connects two distributions in a cost-minimizing manner. Specifically, the source sample $x$ is mapped to the target sample $y=T(x)$ under two constraints: (1) $T(x)$ generates the target distribution $\nu$ and (2) $T$ is chosen to be the cost minimizer $c(x, T(x))$ among various target distribution generators. Importantly, **this framework assumes no additional constraints.**
> > >
> > > In the unpaired setting, the partial point cloud distribution $\mu$ and the target point cloud distribution $\nu$ originate from two different sources. In this case, whether the OT map $T$ becomes a valid point cloud completion model is determined by two factors:
> > > - Similarity between $\nu$ and $\nu_{gt}$
> > > - Appropriateness of the cost function $c(\cdot, \cdot)$ in retrieving the correct point cloud completion, i.e., $y_{gt}(x) \approx y=T(x)$.
> > >
> > > We investigated the second factor in Sec 3.1. **The first factor represents the general assumption for the unpaired point cloud completion models, i.e., $\nu \approx \nu_{gt}$**. Because of this assumption, those models, trained on $\nu$, demonstrate generalizability on $\nu_{gt}$, as in Fig 2 and Table 3 and 4. However, there is still a distribution gap between $\nu$ and $\nu_{gt}$.
> > >
> > > In the UOT framework, we introduce some flexibility in matching the target distribution (Eq. 4). This allows our UOT-UPC model to adapt by avoiding overfitting to target samples from $\nu$, that differ significantly from the source samples from $\mu$. **This flexibility enhances its robustness and performance in addressing distribution mismatches.**
> > >
> > > $ $
> > >
> > > ---
> > > **Q2.**
> > > The results in Fig. 11 and Fig. 12 cannot compare the proposed method with other methods given the same inputs.
> > >
> > > **A.**
> > > Unfortunately, we were unable to provide completion results on the same input, as ACL-SPC did not provide the seed for the input partial point cloud.  As an alternative, **we additionally provided 8 samples of point cloud completion results in Fig 11.** The results are consistent. Our UOT-UPC model achieves higher-fidelity point cloud completion, demonstrating better global structure and more evenly distributed points. In contrast, existing unpaired approaches and ACL-SPC exhibit sparse results.
> > >
> > > $ $
> > >
> > > ---
> > >
> > > We hope these additional responses are helpful in addressing the reviewer's concerns. Additionally, we would like to kindly ask the reviewer to consider additional experimental results on the PCN dataset while evaluating our manuscript.

---

### Official Review · Reviewer_D7gL · 2024-11-01

**Soundness:** 1
**Presentation:** 2
**Contribution:** 2
**Rating:** 3
**Confidence:** 3

**Summary:**

This paper introduces UOT-UPC, an unpaired point cloud completion model based on the unbalanced optimal transport (UOT) map. The key idea is to train a model that learns the UOT map between the distribution of incomplete point clouds and the distribution of complete point clouds. This approach leverages the UOT framework's ability to address the class imbalance problem commonly found in unpaired point cloud completion datasets. The paper also identifies the InfoCD cost function as particularly well-suited for unpaired point cloud completion tasks. Experiments show that using InfoCD leads to better performance compared to other cost functions like l2, L2-Chamfer distance, and one-directional L2-Chamfer distance.

**Strengths:**

1. Novelty: UOT-UPC is the first model to apply the unbalanced optimal transport map to unpaired point cloud completion.

2. Robustness to Class Imbalance: The UOT framework allows UOT-UPC to effectively handle class imbalance. Experiments demonstrated that UOT-UPC maintained consistent performance across various class imbalance ratios, better than other models like USSPA and OT-UPC.

3. Better Performance: UOT-UPC achieves better performance on both single-category and multi-category settings on the dataset proposed by Ma et al. 2023.

**Weaknesses:**

1. Limited Cost Function Exploration: The paper's claim of identifying the "optimal" cost function can be challenged. While the authors compare four different cost functions (l2, L2-Chamfer distance (cdl2), one-directional L2-Chamfer distance (cdl2fwd), and InfoCD), this is a relatively small selection of potential options. Other cost functions might exist that could yield even better performance. The study primarily relies on the ShapeNet dataset for evaluating these cost functions.

2. Dataset Dependence: The experiments primarily focus on a single dataset proposed in Ma et al. 2023, despite the availability of other datasets in the field. This raises concerns about the generalizability of UOT-UPC's performance to other datasets. Evaluating the model on a wider variety of datasets would strengthen the paper's conclusions and provide a more comprehensive understanding of UOT-UPC's capabilities and limitations.

**Questions:**

According to the weakness part, please clarify the choice of cost function and report more results on other datasets. Besides, the source mixture trick seems interesting but not well-explained. Can you give more explanation and/or more insights about this trick?

---

> ### Author Response · Authors · 2024-11-21
> **Response to Reviewer D7gL (1/2)**
>
> We sincerely thank the reviewer for carefully reading our manuscript and providing valuable feedback. Moreover, we appreciate the reviewer for acknowledging that "UOT-UPC is the first UOT map model for unpaired point cloud completion" and that "UOT-UPC effectively handles class imbalance problem". We hope our responses to be helpful in addressing the reviewer's concerns. We highlighted the corresponding revisions in the manuscript in Red.
>
> $ $
>
> ---
> **W1.** Limited Cost Function Exploration: The paper's claim of identifying the "optimal" cost function can be challenged. While the authors compare four different cost functions (l2, L2-Chamfer distance (cdl2), one-directional L2-Chamfer distance (cdl2fwd), and InfoCD), this is a relatively small selection of potential options. Other cost functions might exist that could yield even better performance. The study primarily relies on the ShapeNet dataset for evaluating these cost functions.
>
> **A.** Thank you for the thoughtful comment. Following the reviewer's suggestion, we revised the statement from "identifying the optimal cost function" to "comparing various cost functions and identifying the most suitable cost function" in the revised version of our manuscript. Furthermore, we conducted an **additional ablation study on the cost function using the PCN dataset** to further support that InfoCD is the most effective cost function among the four options. The experimental results are presented below:
>
> - Ablation study on the cost function $c(\cdot, \cdot)$ ($cd^{l 1} \times 10^2$ ($\downarrow$)) on the PCN dataset.
>
> |Cost function| cabinet | sofa | lamp |
> |:---|:---|:---|:---|
> |$l_{2}$ | 19.38 | 17.92 | 17.27 |
> |$cd^{l2}$ | 8.52 | 8.33 | 7.23 |
> |${cd^{l2}}_{fwd}$ | 14.28 | 11.70| 11.76 |
> |InfoCD | **6.41** | **8.08** | **6.79** |
>
> The results are consistent with the original ablation study (Table 6) in our manuscript. InfoCD achieved the best performance, while the L2 distance yielded the worst results. We incorporated this additional experiment into our manuscript, which further supports our cost evaluation analysis (Table 10).
>
>
>
> Moreover, we would like to emphasize that we compared four widely used cost functions for point cloud data [1, 2], except for the Earth Mover's distance [1], which is too costly to apply during every training iteration. As the reviewer suggested, there may be other alternatives, such as introducing an encoder. However, investigating parametric cost functions is beyond the scope of this work. Additionally, while our cost evaluation was conducted on the ShapeNet dataset in our original manuscript, we expanded our experiments to include 10 categories in a single-category setting, as well as additional multi-category experiments.
>
> $ $
>
> [1] Fan, Haoqiang, Hao Su, and Leonidas J. Guibas. "A point set generation network for 3d object reconstruction from a single image.", CVPR 2017.
>
> [2] Lin, Fangzhou, et al. "InfoCD: a contrastive chamfer distance loss for point cloud completion.", NeurIPS 2023.
>
> $ $
>
> ---
>
> **W2.** Dataset Dependence: The experiments primarily focus on a single dataset proposed in Ma et al. 2023, despite the availability of other datasets in the field. This raises concerns about the generalizability of UOT-UPC's performance to other datasets. Evaluating the model on a wider variety of datasets would strengthen the paper's conclusions and provide a more comprehensive understanding of UOT-UPC's capabilities and limitations.
>
> **A.** We appreciate the reviewer for the valuable comment. Following the reviewer's advice, we conducted **additional experiments on the PCN dataset**. Our UOT-UPC model also outperforms previous unpaired approaches on this dataset. These additional results are incorporated into the revised version of our manuscript (Table 9).
>
>
> - **Point cloud completion comparison** on the PCN Dataset, assessed by L1 Chamfer Distance $cd^{l 1} \times 10^2$ ($\downarrow$).
>
> |   |Method | AVG | chair | table | cabinet | sofa | lamp |
> |:---|:---|:---|:---|:---|:---|:---|:---|
> |  Unpaired | ShapeInv | 19.05 | 23.18 | 15.66 | 17.14 | 22.85 | 16.40 |
> |  Unpaired | Unpaired | 14.87 | 12.87 | 8.14 | 14.30 | 18.23 | 20.82 |
> |Unpaired| Cycle4 | 17.60 | 14.25 | 15.73 | 21.06 | 21.54 | 15.40 |
> |  Unpaired | USSPA  | 12.63 | 13.52 | 9.66 | 8.89 | 15.51 | 15.57 |
> | Unpaired  | Ours | **7.92** | **10.22** | **8.11** | **6.41** | **8.08** | **6.79** |
>
> $ $
>
> ---

---

> ### Author Response · Authors · 2024-11-21
> **Response to Reviewer D7gL (2/2)**
>
> **Q.** Besides, the source mixture trick seems interesting but not well-explained. Can you give more explanation and/or more insights about this trick?
>
> **A.** The source mixture trick refers to setting the source distribution as a mixture of $50 \\%$ incomplete and $50 \\%$ complete point clouds (Lines 328-335). As discussed in Lines 511-521 (Sec 5.3),  we introduced this trick to **assist our transport map (generator) in generating the target distribution better.** For the mode of the complete point cloud in the source distribution, the optimal transport map ideally learns the identity mapping. Our intuition is that learning the identity mapping for complete point clouds is easier than learning the completion mapping for incomplete point clouds.
> Note that without the source mixture trick, the transport map is solely trained to perform the completion mapping. In contrast, with the source mixture trick, the transport map is trained to handle both the identity mapping for input complete point clouds and the completion mapping for input incomplete point clouds. **We hypothesize that this approach enables the generator to more effectively learn to produce complete point samples by leveraging the simplicity of the identity mapping task.**
> The ablation study (Table 7) empirically supported our hypothesis. Introducing the source mixture trick consistently improved both metrics, Chamfer distance and F-scores. The improvement was more prominent in the fidelity of the generated point clouds (F-scores).

---

> > ### Author Response · Authors · 2024-11-25
> >
> > We sincerely thank the reviewer for the effort in reviewing our paper. We would greatly appreciate it if the reviewer let us know whether our response was helpful in addressing the reviewer's concerns. If there are additional concerns or questions, please let us know. If our responses were helpful in addressing concerns, we kindly ask the reviewer to consider raising the score.

---

> > > ### Comment · Reviewer_D7gL · 2024-11-26
> > >
> > > I appreciate the authors' response and will keep my score after reviewing the rebuttal.

---

### Official Review · Reviewer_dUov · 2024-11-04

**Soundness:** 3
**Presentation:** 3
**Contribution:** 3
**Rating:** 5
**Confidence:** 5

**Summary:**

This paper an unpaired point cloud completion approach based on the unbalanced optimal transport map. The key idea is to formulate the unpaired point cloud completion task as the optimal transport problem and investigate the optimal cost function for this task, and introduce an unbalanced optimal transport framework for addressing the class imbalance problem. Experimental results show the proposed method achieves state-of-the-art performance in unpaired point cloud completion.

**Strengths:**

1.	I like the 3D shape completion topic, and the pipeline is carefully designed.
2.	The proposed method is evaluated on the dataset proposed in USSPA and performs better than SOTAs, although some important SOTAs are missing.
3.	The paper is clear and easy to follow.

**Weaknesses:**

1.	The authors should test more categories to explore the effects of class imbalance.
2.	The authors should conduct more datasets for demonstrating the effectiveness of the proposed mehtods, such as PCN datasets.
3.	Some important SOTA methods for 3D shape completion are missing. The authors should compare and discuss them with the proposed method.
[1] ASFM-Net: Asymmetrical Siamese Feature Matching Network for Point Completion
[2] 3D Shape Generation and Completion Through Point-Voxel Diffusion
4. The computational cost should be analyzed and compared with the other methods.
5. The authors are encouraged to provide code for reimplementation.

**Questions:**

1.	Compared with the diffusion based mehtods, what are the advantages of the proposed mehtod?
2.	Compared with USSPA, the proposed method has comparable (lower) performance in some categories. What is the reason?
3.	Is it possiable to complete unseen categories?
4.	Do the authors think the CD or F_score are the best metrics for evaluating 3D shape completion?

---

> ### Author Response · Authors · 2024-11-21
> **Response to Reviewer dUov (1/2)**
>
> We sincerely thank the reviewer for carefully reading our manuscript and providing valuable feedback. We hope our responses to be helpful in addressing the reviewer's concerns. We highlighted the corresponding revisions in the manuscript in Blue.
>
> $ $
>
> ---
> **W1.** The authors should test more categories to explore the effects of class imbalance.
>
> **A.** Following the reviewer's advice, we conducted **additional experiments on the class pairs of (lamp, trash bin) and (lamp, bed) under various class imbalance ratios $r$**. As in Table 5, we evaluated both models using L1-Chamfer distance ($cd^{l1} \times 10^{2}$ ($\downarrow$)). Note that in our original manuscript, we conducted experiments on (TV, Table). The experimental results are presented below:
>
> - lamp : trash bin
>
> |r| 0.3 | 0.5 | 0.7 | 1|
> |:---|:---|:---|:---|:---|
> |USSPA | 10.16 | 9.49 | 10.21 | 10.21 |
> |OT | 22.03 | 21.37 | 21.07 | 29.43 |
> |Ours | **9.24** | **9.01** | **9.39** | **9.41** |
>
> $ $
>
> - lamp : bed
>
> |r| 0.3 | 0.5 | 0.7 | 1|
> |:---|:---|:---|:---|:---|
> |USSPA | 9.64 | 9.78 | 9.27 | 9.79 |
> |OT | 22.68 | 20.18 | 22.91 | 22.75|
> |Ours | **8.65** | **8.83** | **8.87** | **9.04** |
>
> As in the original results, our UOT-UPC consistently outperforms USSPA under class imbalance. We incorporated these additional results into the revised version of our manuscript (Table 8). We appreciate the reviewer for the valuable suggestion, which helped strengthen our claim.
>
> $ $
>
> ---
> **W2.** The authors should conduct more datasets for demonstrating the effectiveness of the proposed mehtods, such as PCN datasets.
>
> **A.** We appreciate the reviewer for providing constructive suggestions. Following the reviewer's suggestion, we conducted **additional experiments on the PCN dataset**. Our UOT-UPC model also outperforms previous unpaired approaches on this dataset. These additional results are incorporated into the revised version of our manuscript (Table 9).
>
> - Point cloud completion comparison on the PCN Dataset, assessed by L1 Chamfer Distance $cd^{l 1} \times 10^2$ ($\downarrow$).
>
> |   |Method | AVG | chair | table | cabinet | sofa | lamp |
> |:---|:---|:---|:---|:---|:---|:---|:---|
> |Unpaired| ShapeInv | 19.05 | 23.18 | 15.66 | 17.14 | 22.85 | 16.40 |
> |Unpaired| Unpaired | 14.87 | 12.87 | 8.14 | 14.30 | 18.23 | 20.82 |
> |Unpaired| Cycle4 | 17.60 | 14.25 | 15.73 | 21.06 | 21.54 | 15.40 |
> |Unpaired| USSPA  | 12.63 | 13.52 | 9.66 | 8.89 | 15.51 | 15.57 |
> |Unpaired| Ours | **7.92** | **10.22** | **8.11** | **6.41** | **8.08** | **6.79** |
>
> $ $
>
> ---
> **W3.** Some important SOTA methods for 3D shape completion are missing. The authors should compare and discuss them with the proposed method. [1] ASFM-Net: Asymmetrical Siamese Feature Matching Network for Point Completion [2] 3D Shape Generation and Completion Through Point-Voxel Diffusion
>
> **A.** Thank you for suggesting SOTA methods for 3D shape completion. We additionally cited ASFM-Net and PVD in Line 36. However, **we respectfully believe that these two works are supervised approaches, and therefore not directly comparable to our unsupervised model**. Specifically, ASFM-Net requires paired partial and complete data to train an autoencoder in its first stage of training. PVD utilizes the shape information of partial point clouds as a condition for the diffusion model, which also relies on paired data. As our model is designed for unsupervised point cloud completion, these methods are not appropriate for direct comparison.
>
> $ $
>
> ---
> **W4.** The computational cost should be analyzed and compared with the other methods.
>
> **A.** We compared the training time with the most relevant previous work, USSPA. The point cloud completion cost is the same because we share the same backbone architecture. **The training costs are comparable.** Specifically, the training costs, for the same lamp category and the same 480 epoch, are as follows:
>
> - Train time comparison on lamp
>
> |Train (480 Epoch) | Time (sec) |
> |:---|:---|
> |USSPA | 1103.87 |
> |Ours | 1320.80 |
>
> $ $
>
> ---
> **W5.** The authors are encouraged to provide code for reimplementation.
>
> **A.** In the original submission, we included the source code of our model in the supplementary material.
>
> $ $
>
> ---

---

> ### Author Response · Authors · 2024-11-21
> **Response to Reviewer dUov (2/2)**
>
> **Q1.** Compared with the diffusion based mehtods, what are the advantages of the proposed mehtod?
>
> **A.** First, our model provides **efficient sample generation** with only 1 NFE (Number of Function Evaluations). In contrast, the diffusion models require a larger number of NFEs (usually $\geq 100$), i.e., neural network inference. This makes our UOT-UPC significantly faster for sample generation. Moreover, our UOT-UPC leverages the Unbalanced Optimal Transport (UOT) theory. The UOT theory provides a **theoretical justification for the robustness to class imbalance of our model**. We validated this robustness through experiments (Table 5).
>
> $ $
>
> ---
> **Q2.** Compared with USSPA, the proposed method has comparable (lower) performance in some categories. What is the reason?
>
> **A.** In Table 3, our model outperforms USSPA in 7 out of 10 categories, achieves comparable results in 2 categories, and shows relatively inferior performance in the cabinet category (10.77 vs. 11.84). However, we would like to emphasize that our model outperformed USSPA on Average in Table 3 and also on PCN datasets, as discussed in the response to W2.
>
> Regarding this cabinet category, note that we evaluated our model in all experiments using the same hyperparameter, i.e., across the multi-category setting and all categories in the single-category setting. However, the optimal cost intensity parameter $\tau$ in $c = \tau \cdot \text{InfoCD}$ depends on the data distributions. In the qualitative samples from the cabinet category (Fig 7), we observed that the point cloud completion from our model was more similar to the input partial point cloud, compared to USSPA. In this respect, we suspect that we applied a larger $\tau$ than the optimal value for the cabinet distribution.
>
> $ $
>
> ---
> **Q3.** Is it possible to complete unseen categories?
>
> **A.** We believe that this would be not feasible with our UOT-UPC model, because our model is based on the UOT map between the seen incomplete and complete point cloud data. We think this kind of generalization could be more suited to a Foundation Model trained on a large, diverse dataset. We also find the idea of exploring alternative approaches for point cloud completion on unseen categories interesting and look forward to future research in this direction.
>
> $ $
>
> ---
> **Q4.** Do the authors think the CD or F_score are the best metrics for evaluating 3D shape completion?
>
> **A.**
> Thank you for your insightful question. While the Chamfer Distance (CD) and F-score are commonly used metrics for evaluating 3D point cloud completion, we believe they each have their strengths and limitations. The CD is effective for measuring the closeness of point clouds, as it evaluates how well the generated points match the target points by finding the closest point between them. However, it might not capture certain structural or topological details. For example, when a large portion of points are concentrated together, the CD might not evaluate whether other small portions of details are recovered. The F-score, on the other hand, provides a balance between precision and recall, which can be particularly useful when evaluating how well the model captures both the presence of points and the overall shape. For our work, we chose these metrics because they are widely used in previous works. However, we agree with the reviewer that there is room for development in developing better metrics for 3D shape completion.

---

> > ### Author Response · Authors · 2024-11-25
> >
> > We sincerely thank the reviewer for the effort in reviewing our paper. We would greatly appreciate it if the reviewer let us know whether our response was helpful in addressing the reviewer's concerns. If there are additional concerns or questions, please let us know. If our responses were helpful in addressing concerns, we kindly ask the reviewer to consider raising the score.

---

### Author Response · Authors · 2024-11-25
**General response**

Dear reviewers,

We thank the reviewers for their thoughtful comments and suggestions. We are pleased that the reviewers recognize the novelty of our framework as the first model to utilize the unbalanced optimal transport map [D7gL] and acknowledge its superior performance compared to previous models [dUov, D7gL, uLAt].

We consider that the main concern is that the evaluation of our model primarily relies on the USSPA benchmark (dUov, D7gL, uLAt), although we adopted both the single-category and multi-category scenarios. To address this concern, we have conducted the following additional experiments:

- (1) We evaluated our model's point cloud completion performance on the PCN dataset (**Table 9**) [dUov, D7gL, uLAt].
- (2) We evaluated the generalizability of our model on real scan data (KITTI dataset) (**Fig 11**) [uLAt].
- (3) We evaluated the robustness of our model to the class imbalance problem by testing two different class combinations across diverse class imbalance ratios (**Table 8**) [dUov].
- (4) We demonstrated that our cost function evaluation for the OT framework generalizes to another dataset by conducting an additional cost ablation study on the PCN dataset (**Table 10**) [D7gL].

We hope our responses to be helpful in addressing the reviewer's concerns. We highlighted the corresponding revisions in the manuscript in Blue [dUov], Red [D7gL], and Brown [uLAt].

---

### Meta-Review · Area_Chair_LDYh · 2024-12-19

**Metareview:**

After rebuttal and multiple rounds of discussion, all three reviewers unanimously agreed to reject this submission, especially the problem formulation and inferior performance. The specific comments provided by reviewers are presented as follows.


Reviewer dUov：The paper lacks experiments on diverse datasets (e.g., PCN) and does not explore the effects of class imbalance thoroughly. Additionally, comparisons with key state-of-the-art methods (e.g., ASFM-Net and Point-Voxel Diffusion) and analysis of computational costs are missing.

Reviewer D7gL: The exploration of cost functions is limited, and the claim of identifying the "optimal" cost function is overstated. Experiments rely heavily on a single dataset (USSPA), raising concerns about generalizability, and additional datasets like PCN or KITTI should be used to validate the method further.

Reviewer uLAt: The assumption of distribution alignment between incomplete and complete point clouds is unrealistic in many cases, as datasets like ShapeNet and ScanNet come from different sources. Experiments are limited to Scan2CAD, and comparisons with more relevant methods (e.g., ACL-SPC and CloudMix) are lacking.


Two reviewers recommended rejection (scores of 3), and one provided a marginally below acceptance threshold score (5). So I recommend “reject”.

**Additional Comments On Reviewer Discussion:**

Two reviewers provided "reject" and one provided "marginally below".

---

### Decision · Program_Chairs · 2025-01-22

Reject